# A protein kinase coordinates cycles of autophagy and glutaminolysis in invasive hyphae of the fungus *Magnaporthe oryzae* within rice cells

Gang Li[1,5], Ziwen Gong[1,2,5], Nawaraj Dulal [1], Margarita Marroquin-Guzman[1,3], Raquel O. Rocha [1,4], Michael Richter[1] & Richard A. Wilson [1] ✉

The blast fungus *Magnaporthe oryzae* produces invasive hyphae in living rice cells during early infection, separated from the host cytoplasm by plant-derived interfacial membranes. However, the mechanisms underpinning this intracellular biotrophic growth phase are poorly understood. Here, we show that the *M. oryzae* serine/threonine protein kinase Rim15 promotes biotrophic growth by coordinating cycles of autophagy and glutaminolysis in invasive hyphae. Alongside inducing autophagy, Rim15 phosphorylates NAD-dependent glutamate dehydrogenase, resulting in increased levels of α-ketoglutarate that reactivate target-of-rapamycin (TOR) kinase signaling, which inhibits autophagy. Deleting *RIM15* attenuates invasive hyphal growth and triggers plant immunity; exogenous addition of α-ketoglutarate prevents these effects, while glucose addition only suppresses host defenses. Our results indicate that Rim15-dependent cycles of autophagic flux liberate α-ketoglutarate – via glutaminolysis – to reactivate TOR signaling and fuel biotrophic growth while conserving glucose for antioxidation-mediated host innate immunity suppression.

*Magnaporthe oryzae* (syn *Pyricularia oryzae*)[1] causes blast, the most devastating disease of cultivated rice[2–4]. *M. oryzae* is a hemibiotroph, an important class of eukaryotic microbial plant pathogen that, following host invasion, elaborate invasive hyphae (IH) that grow undetected for an extended period in living host plant cells before switching from this biotrophic growth phase to necrotrophy, when host cells die and disease symptoms develop. In *M. oryzae*, IH are wrapped in the plant-derived extra-invasive hyphal membrane (EIHM), forming an interfacial compartment into which apoplastic effectors like Bas4 can be deployed[5,6]. *M. oryzae* IH also associate with a focal plant lipid-rich structure called the biotrophic interfacial complex (BIC). A single BIC forms in the first infected rice cell following penetration at around 28–32 h post inoculation (hpi) and at the tips of IH as they spread into neighboring cells at around 44 hpi. BICs form outside IH and accumulate secreted cytoplasmic effectors such as Pwl2 before they are translocated into the host cell[5,6]. Secreted effector proteins act to evade host detection or suppress plant defense responses. In addition to effectors, *M. oryzae* antioxidation systems are also essential to the success of biotrophic invasion by scavenging host reactive oxygen species (ROS) that otherwise trigger plant innate immunity[7–10].

We seek to understand the metabolic strategies and molecular decision-making processes employed by *M. oryzae* to ensure metabolic

[1]Department of Plant Pathology, University of Nebraska-Lincoln, Lincoln, NE, USA. [2]State Key Laboratory for Biology of Plant Diseases and Insect Pests, Institute of Plant Protection, Chinese Academy of Agricultural Sciences, Beijing, China. [3]Present address: Bayer CropScience, Chesterfield, MO, USA. [4]Present address: Department of Plant Pathology and Ecology, The Connecticut Agricultural Experiment Station, New Haven, CT, USA. [5]These authors contributed equally: Gang Li, Ziwen Gong. ✉e-mail: rwilson10@unl.edu

homeostasis during biotrophy, when resources must be allocated between IH growth, maintaining interfacial integrity, and plant defense suppression. How fungal metabolic homeostasis is achieved during biotrophy is particularly intriguing when considering that genetic studies indicate fungal acquisition of host purines and at least some amino acids is severely limited or non-existent during biotrophy[11–13]. Previously, we uncovered a role for the *M. oryzae* target-of-rapamycin (TOR)-Imp1-autophagy signaling axis in mediating biotrophic growth and maintaining biotrophic interface integrity[14]. TOR kinase is a conserved signaling hub that promotes growth and suppresses autophagy in the presence of nutrients and energy[15]. Imp1-dependent autophagy in *M. oryzae* IH ensures biotrophic interfacial membrane integrity by maintaining membrane homeostasis; loss of *IMP1* led to biotrophic interfacial membrane erosion and leakage of effectors into host cells[14]. However, despite uncovering the importance of *M. oryzae* autophagy to biotrophic interfacial membrane integrity, the precise metabolic role(s) of autophagy during biotrophic growth are unknown.

To gain more insights into the role of autophagy and its regulation on biotrophic growth, we targeted the *M. oryzae* RIM15 homolog encoding a serine/threonine protein kinase. In yeast, Rim15 functions to integrate signaling from the nutrient-sensing kinases TORC1, Sch9, PKA and Pho85-Pho80. It is required for starvation-induced autophagy in response to PKA and Sch9 inactivation, but yeast Rim15 is not required for TORC1-dependent rapamycin-induced autophagy, suggesting ScRim15 regulates autophagy in parallel to the TORC1 pathway[16–19]. We hypothesized *M. oryzae* Rim15 played a similar role in TOR-independent autophagy regulation that may impact biotrophy.

Here, by examining the Δ*rim15* deletion mutant strain using a combination of live-cell imaging, plate tests and multiomic approaches, we show that biotrophic growth requires cycles of Rim15-coordinated autophagy and glutaminolysis to yield α-ketoglutarate, a cell-intrinsic nutrient sufficiency signal that reactivates TOR and fuels growth while conserving glucose for antioxidation-mediated host defense suppression. Our results provide a mechanistic framework for understanding how fungal metabolism is integrated with host plant cell colonization and innate immunity suppression at the molecular, cellular and biochemical level.

## Results

### The serine/threonine protein kinase Rim15 is required for biotrophic growth, maintaining biotrophic interfacial membrane integrity, and plant defense suppression

We disrupted the *M. oryzae* RIM15 coding region (MGG_00345)[20] in our WT Guy11 isolate using homologous gene recombination to replace the first 1 kb of the *RIM15* coding sequence with the *ILV2* gene conferring sulphonylurea resistance[14,21]. More than ten Δ*rim15*-carrying mutant strains were identified by PCR and two deletants, Δ*rim15* #7 and #9, were initially characterized and found to be indistinguishable (Fig. 1a,b and Supplementary Fig. 1a-d). Δ*rim15* #7 was used as the recipient for the plasmid carrying the *RIM15-GFP* gene to generate the Δ*rim15 RIM15-GFP* complementation strain (Fig. 1a). Δ*rim15* #7 was also the recipient of the pBV591 vector[5] to generate a Δ*rim15* mutant strain producing BIC-localized Pwl2-mCherry:NLS and apoplast localized Bas4-GFP (Fig. 1c). Compared to the WT parental strain, Δ*rim15* #7 and #9 deletion strains were marginally reduced in radial diameter on nutrient-rich complete media (CM), and both were reduced for sporulation on CM (Supplementary Fig. 1a,b). Sporulation was improved on oatmeal agar, enabling enough Δ*rim15* spores to be harvested for downstream applications. Applying equal numbers of spores to 3-week-old seedlings of the susceptible CO-39 rice cultivar showed that compared to WT and the complementation strain, Δ*rim15* #7 and #9 mutant strains were non-pathogenic, with both deletants producing small, Type I lesions[22] that did not yield spores (Fig. 1a). Thus, *RIM15* is required for rice blast disease.

Δ*rim15* #7 and #9 formed normal-looking appressoria that were melanized and inflated on artificial hydrophobic surfaces by 24 hpi (Supplementary Fig. 1c). Δ*rim15* #7 and #9 appressoria formed at rates that were indistinguishable from WT on both hydrophobic surfaces and detached rice leaf sheath surfaces, and Δ*rim15* appressoria penetrated rice leaf sheaths at rates comparable to WT (Supplementary Fig. 1d). However, although able to elaborate IH in the first infected rice epidermal cells, subsequent biotrophic growth was curtailed compared to WT at 28 hpi (Supplementary Fig. 1e) and 38 hpi (Fig. 1b), and by 44 hpi, Δ*rim15* #7 and #9 IH movement into adjacent cells was rarely observed (Supplementary Fig. 1d). Thus, *RIM15* is required for establishing extensive biotrophic growth in the first infected rice cell.

We next asked whether reduced biotrophic growth by Δ*rim15* strains impacted interfacial membrane integrity. Using strains expressing *Pwl2-mCherry:NLS* and *BAS4-GFP*, Fig. 1c shows how by 36 hpi, in both Δ*rim15* and the control *RIM15*+ strain, Pwl2-mCherry:NLS accumulated in the BIC and Bas4-GFP accumulated in the apoplastic space to outline IH, as previously described[5,6,8,14]. However, about 4% of Δ*rim15*-infected rice cells displayed some Bas4-GFP leakage into the host cytoplasm. By 39 hpi, when all WT IH carried BICs and intact apoplastic spaces, over half of Δ*rim15*-infected rice cells (*n* = 50, repeated in triplicate) had no visible Pwl2-mCherry:NLS in BICs and Bas4-GFP was in the rice cytoplasm, indicating BIC and EIHM integrity was lost (Fig. 1d). The erosion of biotrophic interfaces was similarly observed previously in the autophagy-defective Δ*imp1* strains[14], albeit at much earlier timepoints (after 28 hpi and before 32 hpi for Δ*imp1*). By 44 hpi, all Δ*rim15*-infected rice cells had lost their BICs and Bas4-GFP was observed in rice cytoplasm, but this was often obscured by robust plant responses that were visible as deposits in host cells (Fig. 1e). These deposits were accompanied by the accumulation of host ROS (Supplementary Fig. 1f), together indicating activated host defenses[9,10] in Δ*rim15*-infected cells. Such strong host cell defense responses were not observed for *RIM15*+-infected rice cells and had not been observed previously for Δ*imp1*-infected rice cells[14]. When considered together, these first results showed that *RIM15* was not required for establishing IH growth and biotrophic interfacial membrane integrity but was, like *IMP1*, required for maintaining these processes. Furthermore, our results show *RIM15* plays an additional role in preventing visible host defense responses to infection after 39 hpi.

To confirm that Bas4-GFP observed in the cytoplasm of Δ*rim15*-infected rice cells resulted from the loss of biotrophic interfacial membrane integrity and was not due to some other phenomena such as active secretion of Bas4-GFP into the rice cell, we treated WT- and Δ*rim15*-infected rice cells with Trypan Blue, an azo dye unable to penetrate cells with intact membranes. Figure 2a shows how, at 39 hpi and 44 hpi, WT infected cells were unstained, indicating intact biotrophic interfacial membrane integrity. In contrast, Δ*rim15*-infected rice cells were stained with Trypan Blue in proportions that were similar to those seen for cells exhibiting Bas4-GFP leakage into the rice cytoplasm in Fig. 1d,e. Thus, Bas4 leakage into host cytoplasm is most likely due to the loss of biotrophic interfacial membrane integrity in Δ*rim15*-infected rice cells.

Next, we sought to determine whether impaired biotrophic growth and loss of interfacial membrane integrity was a consequence of the elevated host defenses in Δ*rim15*-infected rice cells. Figure 2b shows how killing rice leaf sheath cells with ethanol abolished plant defense responses in Δ*rim15*-infected cells but did not remediate Δ*rim15* biotrophic growth. Similarly, Fig. 2c shows how treatment with the NADPH oxidase (NOX) inhibitor diphenyleneiodonium chloride (DPI), which suppresses the host oxidative burst that otherwise triggers plant innate immunity responses[10], prevented host defense responses in Δ*rim15*-infected rice cells but did not remediate Δ*rim15* biotrophic growth or prevent the loss of Δ*rim15* biotrophic interfacial integrity. Thus, the growth defects of Δ*rim15* IH result from the loss of *RIM15* function and are not due to exposure to elevated host defenses.

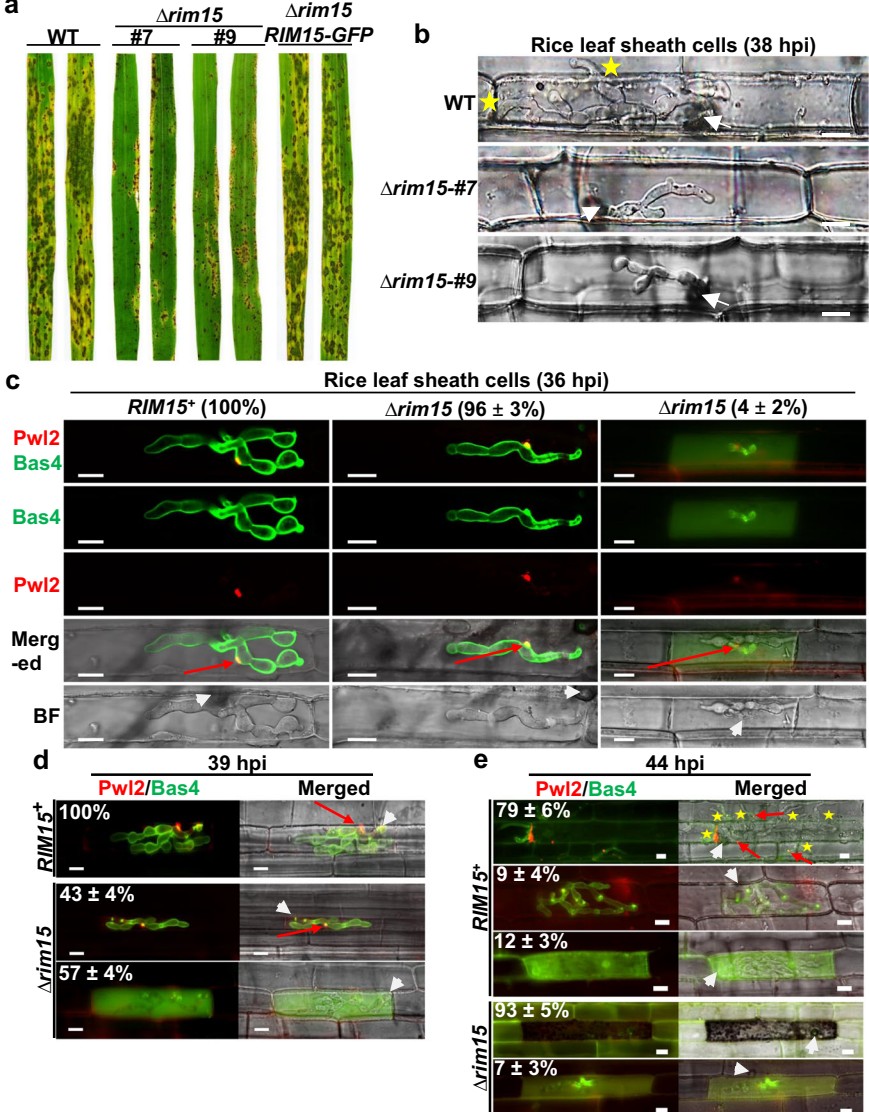

**Fig. 1 | *M. oryzae RIM15* is required for biotrophic growth and interfacial membrane integrity. a** Rice blast disease symptoms of leaves infected with the indicated strains. Spores were applied to 3-week-old rice seedlings of the susceptible cultivar CO-39 at a rate of $1 \times 10^5$ spores ml$^{-1}$. Images were taken at 120 h post inoculation (hpi). **b** Live-cell imaging at 38 hpi of detached rice leaf sheath epidermal cells infected with the indicated strains showing how Δ*rim15* strains are impaired for biotrophic growth. **c–e** Live-cell imaging at 36 hpi (**c**), 39 hpi (**d**) and 44 hpi (**e**) of detached rice leaf sheath epidermal cells infected with the indicated

strains expressing Pwl2-mCherry:NLS and Bas4-GFP as membrane integrity markers. White arrowheads indicate appressorial penetration sites. Red arrows indicate BICs in the merged channel for ease of viewing. Stars indicate movement of IH into neighboring cells. BF is brightfield. Scale bars are 10 μm. Representative images and values are derived from observing 50 infected rice cells per leaf sheath per strain. *n* = 3 biological replicates. Values are means ± SD. **c–e** Source data are provided as a Source Data file.

Rapamycin treatment, which induces autophagy via TOR signaling, remediated biotrophic interfacial integrity in Δ*rim15*-infected rice cells (although biotrophic growth was attenuated for both *RIM15*⁺ and Δ*rim15* IH when TOR was constitutively inactivated by rapamycin) (Fig. 2d). This contrasts with Δ*imp1*, which was not responsive to rapamycin[14] and suggests, like in yeast, that *RIM15* induces autophagy in parallel to TORC1-dependent rapamycin-induced autophagy. Poor growth of Δ*rim15* strains on starvation water agar media compared to WT confirms a role for *RIM15* in autophagy activation under nutrient starvation conditions (Fig. 2e). Considered together, our results suggest that *M. oryzae RIM15* acts on autophagy in a pathway parallel to TOR-Imp1-autophagy signaling to maintain (but not establish) biotrophic interfacial membrane integrity and early biotrophic growth while, at later timepoints, acting independently of the TOR-Imp1 pathway to (directly or indirectly) inhibit host innate immunity induction (Fig. 2f).

By acting on autophagy in parallel to TOR-rapamycin signaling, *M. oryzae RIM15* is functionally similar to yeast *RIM15*. Yeast Rim15p translocates to the nucleus under nutrient starvation and other stress conditions[19,23,24]. However in contrast, and although the *M. oryzae RIM15-GFP* gene complements the Δ*rim15* genetic lesion (Fig. 1a and Supplementary Fig. 2a), the Rim15-GFP protein was cytoplasmic under all tested conditions (Supplementary Fig. 2b–d). Differences in localization of the two Rim15 proteins may reflect evolutionary changes in response to the different lifestyles of these two widely separated fungi, including changes to major downstream targets, as suggested below.

### Rim15 is required for autophagy induction and autophagic flux cycling *in planta*

To further elaborate the role of *RIM15* in biotrophic growth, we next sought to better understand how autophagy was affected in Δ*rim15*

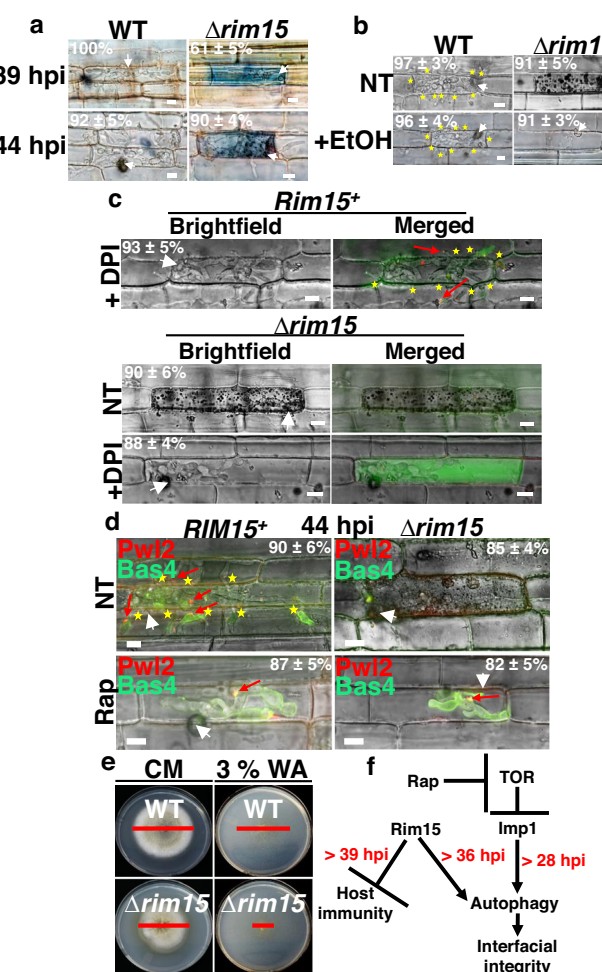

**Fig. 2 | RIM15 acts on autophagy and plant defense suppression in parallel to TOR signaling. a** Images of detached rice leaf sheath epidermal cells infected with the indicated strains and stained with 0.4% Trypan Blue to distinguish infected plant cells with intact membranes (clear) from those with ruptured membranes (blue). **b** Images at 42 hpi of detached leaf sheath epidermal cells infected with the indicated strains following rice cell killing by treatment with 30% ethanol (EtOH). **c**, **d** Live-cell imaging of detached rice leaf sheath epidermal cells infected with the indicated strains expressing Pwl2-mCherry:NLS and Bas4-GFP. **c** Spore suspensions were supplemented with 0.4 μM of the NOX inhibitor diphenyleneiodonium (DPI) prior to inoculation to suppress the host ROS burst. Cells were imaged at 42 hpi. **d** 10 μM rapamycin (Rap) dissolved in 1% DMSO was added to infected rice cells at 36 hpi and cells were imaged at 44 hpi. **e** Plate tests showing impaired growth of the Δ*rim15* mutant strain on 3% water agar (WA) media without nutrients, compared to WT. Red bar indicates colony diameters. CM is complete media. Images were taken at 12 dpi. **f** Model summarizing relationship of Rim15 to plant defense suppression, autophagy and the TOR-Imp1 signaling pathway[14]. > denotes protein acts on the indicated process after the given timepoint. *RIM15* is required for optimal growth by 28 hpi, but not for interfacial membrane integrity until after 36 hpi or for plant defense suppression until after 39 hpi. **a**–**d** White arrowheads indicate appressorial penetration sites. Red arrows indicate BICs in the merged channel for ease of viewing. Stars indicate movement of IH into neighboring cells. Scale bars are 10 μm. Representative images and values are derived from observing 50 infected rice cells per leaf sheath per strain. *n* = 3 biological replicates. Values are means ± SD. **b**–**d** NT is no treatment. **a**–**e** Source data (including colony diameter statistics) are provided as a Source Data file.

strains. To achieve this, we transformed into WT and Δ*rim15* #7 proto-plasts a plasmid encoding the full-length *M. oryzae* autophagy-related protein 8 (Atg8), fused to GFP at its N-terminus, in order to perform the GFP-Atg8 processing assay for bulk autophagic activity[17,19,25] (Fig. 3a). For each strain carrying *GFP-ATG8*, two independent transformants

were analyzed and found to have identical phenotypes with regards to GFP-Atg8 localization. Figure 3b shows that in *RIM15*+, as expected, little or no autophagic activity was detected in vegetative mycelia in nutrient-rich complete media (CM), with GFP-ATG8 present in the cytoplasm and absent from the vacuole, whereas under nutrient-starvation conditions (ie. in $H_2O$), GFP accumulated in vacuoles, indicating increased autophagic flux. In contrast, Δ*rim15* did not accumulate GFP in vacuoles in either CM or water, with GFP-Atg8 observed in mycelial cytoplasm under both conditions, indicating that Δ*rim15* strains were blocked in autophagy induction and that, like in yeast[17], *M. oryzae* Rim15 is required for autophagic flux in response to nutrient starvation.

*In planta*, autophagic activity (as measured by the percentage of vacuoles containing GFP) was induced in *RIM15*+ at earlier timepoints than expected (28 hpi and 36 hpi), and at 44 hpi during cell-to-cell movement, which is consistent with our previous results that autophagy activation stimulates cell-to-cell movement[14]. However, autophagy was unexpectedly diminished in WT at 32 hpi and, more prominently, at 40 hpi, when GFP-ATG8 was instead predominant in cytoplasm (Fig. 3c–g). In contrast, GFP in vacuoles was not observed in Δ*rim15* IH until 40 hpi. We conclude that autophagic flux in *M. oryzae* IH during biotrophic growth is cyclical and this cycling is abolished in Δ*rim15* (Fig. 3h, solid lines).

The late autophagy observed at 40 hpi in Δ*rim15* IH—which is *RIM15*-independent and, considering the accompanying plant ROS burst, could be the result of oxidative stress-activated autophagy or nonapoptotic autophagic cell death in response to stress[25,26]—was insufficient to rescue the loss of biotrophic growth and membrane integrity in Δ*rim15* strains, leading us to hypothesize that early autophagy and autophagic cycling was necessary for biotrophic growth. In support of this notion, Supplementary Fig. 3 shows how adding the autophagy inhibitor 3-methyladenine (3-MA) to rice leaf sheaths infected with WT up to 46 hpi abolished further biotrophic growth, presumably by inhibiting the next round of autophagic induction. Furthermore, in order to confirm that the GFP measured in vacuoles was due to active autophagic flux and was not the result of nonspecific turnover of GFP-ATG8, we added 3-MA at the two troughs of autophagy activity preceding autophagy induction in WT with the expectation that this would abolish GFP accumulation in vacuoles. Figure 3i (quantified in Fig. 3h, dashed lines) shows how 3-MA treatment at 32 hpi or 40 hpi abolished the succeeding peaks of GFP accumulation in vacuoles at 36 hpi and 44 hpi, respectively, that were otherwise observed in untreated controls, and inhibited cell-to-cell movement at 44 hpi. These results together confirm that GFP accumulation in untreated vacuoles at 36 hpi and 44 hpi corresponds to induced autophagic activity, and that this activity is required for biotrophic growth. We conclude that *RIM15*-dependent autophagic cycling is essential for biotrophy. Note, however, that *RIM15* is required for optimal growth by 28 hpi, but it is not required for maintaining interfacial membrane integrity in the majority of infected host rice cells until after 36 hpi (Fig. 1b,d), suggesting that basal autophagy in Δ*rim15* (Fig. 3h) is sufficient to maintain biotrophic interfacial membrane integrity (though not support extensive biotrophic growth) prior to 36 hpi.

## Rim15-dependent glutaminolysis is required for host infection

Our next breakthrough in understanding the role of *RIM15* in host infection came from plate test studies, which showed how the Δ*rim15* mutant strain was impaired (although not abolished) for glutaminolysis, as evidenced by reduced and morphologically distinct colony radial growth compared to WT on media containing glutamine or glutamate as sole carbon and nitrogen sources, and when compared to Δ*rim15* growth on glucose-containing media with glutamine or glutamate as sole nitrogen sources (Fig. 4a). Note that, as shown for growth on CM, Δ*rim15* was somewhat reduced in growth on all media tested compared to WT, although in the presence of glucose, colonies were morphologically similar to WT. Anaplerotic glutaminolysis is the use of glutamine

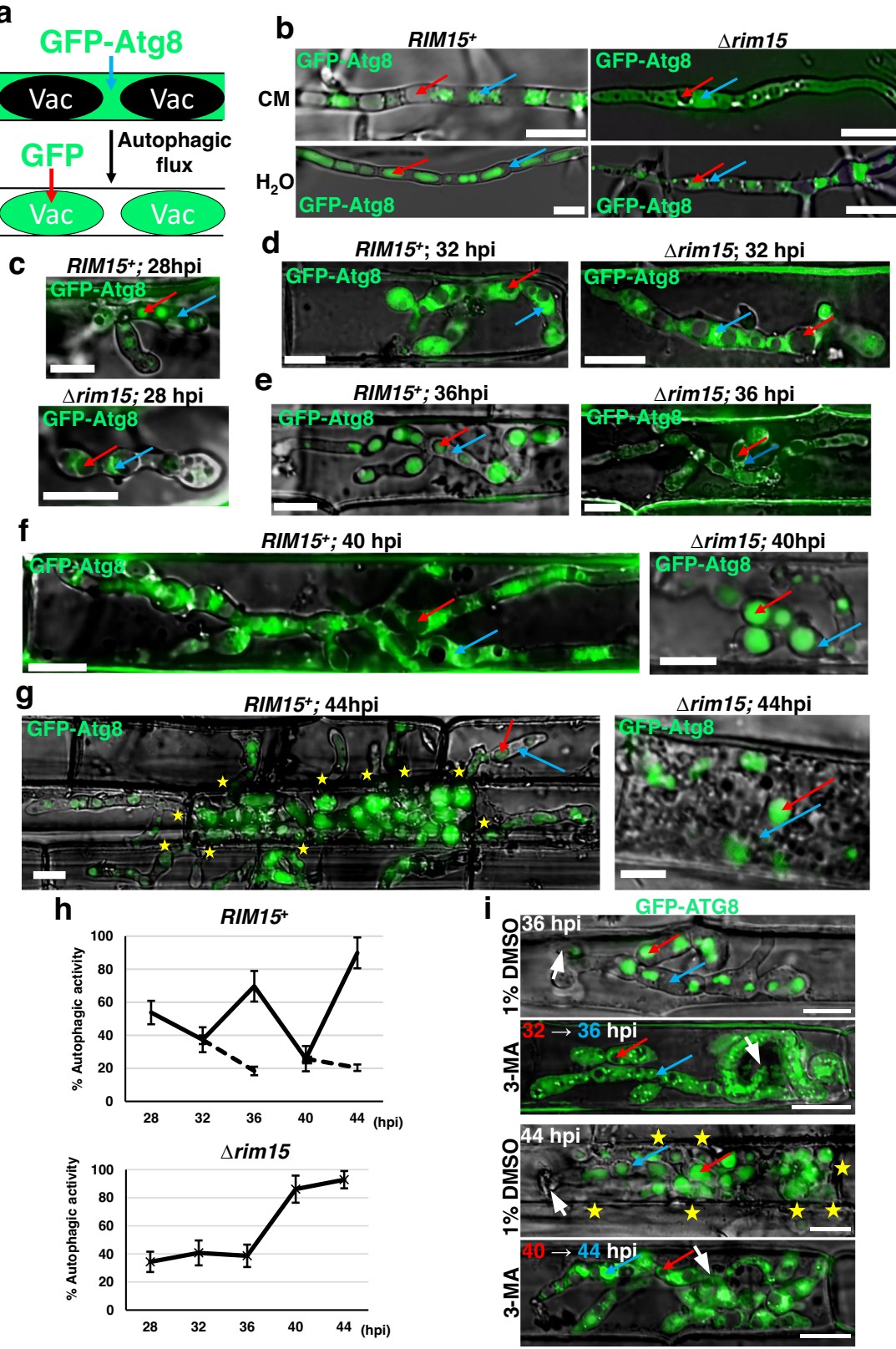

as a carbon source in the absence of glucose via its deamination first to glutamate and then to α-ketoglutarate, which replenishes TCA cycle intermediates (Fig. 4b). To determine how the Rim15 kinase protein might influence glutaminolysis, and whether glutaminolysis was required for infection, we performed proteomic (Supplementary Data 1) and phosphoproteomic (Supplementary Data 2) analyses on three

independent WT and Δ*rim15* mycelial samples following growth in liquid minimal media containing 0.5 mM glutamine as the sole carbon and nitrogen source. Focusing on protein abundance in WT under this growth condition, we detected, as expected, Gdh2/Mgd1 (MGG_05247; UniProt identifier: G4N5E0), the NAD-specific glutamate dehydrogenase that deaminates glutamate to α-ketoglutarate (Fig. 4b),

**Fig. 3 | *RIM15*-dependent autophagic flux cycling is required for biotrophic growth. a** Schematic of the GFP-Atg8 bulk autophagic activity assay. Autophagy activity results in autophagic bodies carrying GFP-Atg8 fusing with the vacuole, where Atg8 is degraded following lysis of the autophagic body, but GFP is more resistant to proteolysis and thus accumulates. Vac are vacuoles. **b–g, i** Micrographs showing autophagy activity in the indicated strains. Examples of vacuoles are indicated with red arrows; examples of cytoplasm are indicated with blue arrows. Scale bars are 10 μm. Merged channel is shown. **b** Mycelia of the indicated strains expressing GFP-Atg8 were inoculated in liquid CM medium for 42 h before harvesting. Mycelia was washed with water and inoculated into fresh liquid CM or water, as indicated, for 3.5 h before imaging. **c–g, i** Live-cell imaging of detached rice leaf sheath epidermal cells infected with strains expressing GFP-Atg8 at the indicated time points. White arrowheads indicate appressorial penetration sites. Stars indicate movement of IH into neighboring cells. Representative images derived from observing 50 infected rice cells per leaf sheath per strain. $n = 3$ biological replicates. **h** Autophagic activity rates in *Rim15*⁺ and Δ*rim15* IH at the indicated timepoints. Solid lines are values from untreated samples. Dashed lines indicate autophagic activity rates following 3-MA treatment at 32 hpi or 40 hpi. Autophagic activity rates are calculated as the percentage of IH vacuoles accumulating GFP in 50 infected rice cells per leaf sheath per strain. $n = 3$ biological replicates. Values are means ± SD. Source data are provided as a Source Data file. **i** 5 mM 3-methyladenine (3-MA) in 1% DMSO was added to infected detached rice leaf sheath epidermal cells at 32 hpi or 40 hpi and samples were imaged at 36 hpi or 44 hpi, respectively, as indicated. Controls were treated with 1% DMSO.

at high normalized abundances (Supplementary Data 1), whereas we detected Gdh1 (MGG_08074; UniProt identifier: G4MXT4), the NADP-specific glutamate dehydrogenase required for the reverse reaction during ammonium assimilation (Fig. 4b), at normalized abundances that were 472-fold lower than for Gdh2 (Supplementary Data 1). We detected a putative glutaminase (MGG_07512; UniProt identifier: G4N1P6) that might catalyze the first step of glutaminolysis, but it was 289-fold less abundant (Supplementary Data 1) than Glt1 (MGG_07187; UniProt identifier: G4MTM7) encoding glutamate synthase, an enzyme that also converts glutamine to glutamate (Fig. 4b). Glt1 abundance was, moreover, similar to that of Gdh2. Gdh2 protein abundances were similar in both WT and Δ*rim15* samples (Supplementary Data 1), but the comparative phosphoprotein data (Supplementary Data 2) showed a reduction in Gdh2 phosphorylation levels in Δ*rim15* mycelia compared to WT (phosphorylated Glt1 peptides were not detected in either sample). Specifically, peptides with phosphorylated serine at position 24 of the Gdh2 protein sequence were detected with high confidence, based on *P* values, in both WT and Δ*rim15* samples after enrichment, but were 3.4-fold more abundant in WT than Δ*rim15*, suggesting at least partial Rim15-dependent phosphorylation of Gdh2 at this Ser24 residue (phosphorylated threonine residues were also detected in Gdh2, but with less confidence (Supplementary Data 2)). Notably, Ser24 in *M. oryzae* Gdh2 is followed by Pro25 (Supplementary Data 2), which matches the (S/T)P consensus motif targeted by yeast Rim15[27].

Based on the above data, we deleted *GLT1*, *GDH1*, and *GDH2* in WT by homologous recombination. Several independent deletants were assessed for each gene, and one each was used for complementation. Δ*glt1*, like Δ*rim15*, was severely impaired (but not abolished) for growth on glutamine as a sole carbon and nitrogen source—suggesting it is a major but not sole generator of glutamate from glutamine—but it grew like WT on glutamate as a carbon source, and on glutamine and glutamate as nitrogen sources (Fig. 4a). Δ*gdh2* was abolished for growth on glutamine and glutamate as sole carbon sources but grew on media with glutamine and glutamate as nitrogen sources (Fig. 4a), although Δ*gdh2* colonies were morphologically distinct from WT on all permissible growth media. In contrast to Δ*glt1* and Δ*gdh2*, Δ*gdh1* was dispensable for glutaminolysis and grew like WT on glutamine and glutamate media. It was, however, impaired for growth on media with ammonium as the nitrogen source, compared to WT (Supplementary Fig. 4). Figure 4c shows how *GLT1* and *GDH2*, but not *GDH1*, are required for host infection. Δ*glt1* spores could form appressoria and penetrate host cells but IH were impaired in biotrophic growth compared to WT. The Δ*gdh2* mutant strain, in our hands, did not make spores. Nonetheless, unlike WT, mycelial blocks of this mutant failed to colonize wounded rice leaves and did not form lesions. In contrast, Δ*gdh1* spores were fully pathogenic. Thus, genes involved in glutaminolysis, but not the reverse reaction, are required for colonizing host tissue, suggesting glutamine is the preferred carbon source during biotrophy.

We next asked whether phosphorylation of the Ser24 site on Gdh2 had physiological relevance, and whether Ser24 was a downstream target for Rim15. Pathogenicity (Fig. 4c) and glutaminolysis on plates (Fig. 4d) were restored to Δ*gdh2* by complementation with the native *GDH2* gene sequence. However, complementation with a mutated *GDH2* copy encoding a version of Gdh2 carrying the phosphorylation-null amino acid substitution S24A did not remediate infection (Fig. 4c), and growth was not fully restored on glutamine media (Fig. 4d), indicating Ser24 phosphorylation is required for optimal Gdh2 function *in planta*. In contrast to the phospho-null Gdh2 data, expressing in Δ*rim15* a phospho-mimicking version of Gdh2 carrying the substitution S24D almost fully remediated pathogenicity on rice leaves (Fig. 4c), and fully remediated growth on media with glutamine or glutamate as the sole carbon source (Fig. 4d), compared to WT and the Δ*rim15* parental strain (Fig. 4d). Three independent transformants for each strain carrying Δ*gdh2 GDH2*, Δ*gdh2 GDH2^S24A* and Δ*rim15 GDH2^S24D* were tested and found to be indistinguishable. We conclude that Rim15 mediates Gdh2 phosphorylation to control glutaminolysis, a process required for rice infection (Fig. 4e). Furthermore, remediation of Δ*rim15* by phospho-mimicking Gdh2^S24D suggests Gdh2 is a major downstream target of Rim15. This may explain the cytoplasmic localization of Rim15 in *M. oryzae* because according to PSORTII, *M. oryzae* Gdh2 does not localize to the nucleus and likely resides in the cytoplasm.

## α-ketoglutarate from glutaminolysis triggers biotrophic growth

Phospho-mimetic Gdh2^S24D production restored Δ*rim15* infection, leading us to consider that defects in glutaminolytic metabolite abundances in Δ*rim15* impaired biotrophic growth. In support of this notion, metabolomic data showed α-ketoglutarate levels were reduced by almost half in Δ*rim15* mycelia compared to WT following growth in minimal media with glutamine as the sole carbon source (Fig. 5a; Supplementary Data 3). We hypothesized that restoration of α-ketoglutarate levels would remediate Δ*rim15*, and to test this we treated Δ*rim15*-infected detached rice leaf sheath epidermal cells at 36 hpi with 10 mM α-ketoglutarate (as the cell-permeable dimethyl α-ketoglutarate (DMKG) analog). Figure 5b shows how this treatment completely remediated the Δ*rim15* mutant phenotype by 44 hpi. Compared to the untreated Δ*rim15* control, treatment with α-ketoglutarate promoted Δ*rim15* IH cell-to-cell movement and suppressed host defense responses. Biotrophic interfacial integrity was also restored as Δ*rim15* IH in adjacent host cells were outlined with Bas4, indicating an intact EIHM, and new BICs formed at IH tips. Δ*rim15* IH development and growth following α-ketoglutarate treatment was indistinguishable at 44 hpi from α-ketoglutarate-treated *RIM15*⁺ IH, which was itself indistinguishable from *RIM15*⁺ IH growth and development in untreated rice cells (Fig. 5b). Figure 5c shows how similarly, treatments with the glutaminolytic metabolites 10 mM glutamine or 10 mM glutamate also remediated Δ*rim15* biotrophy, but treatment with 10 mM ammonium (NH₄⁺) did not. Considering Δ*rim15* is impaired but not abolished for glutaminolysis, these results suggest excess exogenous glutamine or glutamate might remediate Δ*rim15* biotrophy through conversion to α-ketoglutarate. In contrast, NH₄⁺ assimilation does not contribute to α-ketoglutarate production. Remediation by exogenous glutaminolytic metabolites was dose-dependent, such that

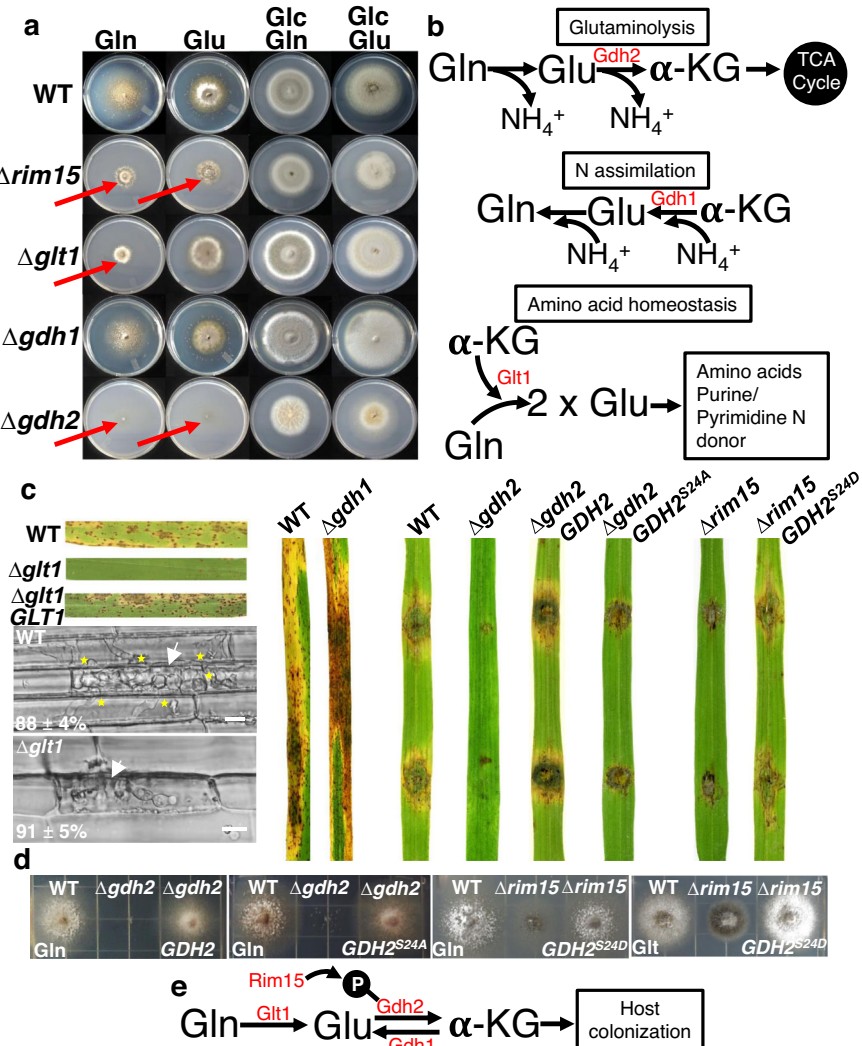

**Fig. 4 | Rim15-controlled glutaminolysis is required for host rice colonization.**
**a** Plate tests showing growth of strains on minimal media containing the indicated carbon and nitrogen sources. Carbon sources were added at a rate of 1% (w/v). Nitrogen sources were at 10 mM final concentration. Images were taken at 10 days and are representative of at least three independent plates per treatment. For ease of viewing, red arrows indicate morphologically distinct, growth reduced, or absent colonies compared to WT on Gln and Glu media without glucose. **b** Schematics of glutamine metabolism during the utilization of glutamine as a carbon source via glutaminolysis and anaplerosis in the TCA cycle, during nitrogen assimilation, and during amino acid homeostasis. Proteins identified in the proteomics data and functionally characterized are shown in red. **c** Glutaminolysis but not nitrogen assimilation is required for rice infection. Rice blast disease symptoms of leaves infected with the indicated strains are shown. *GDH2^{S24A}* encodes a phospho-null version of Gdh2. *GDH2^{S24D}* encodes a phospho-mimicking version of Gdh2. For spray assays, spores were applied to 3-week-old rice seedlings of the susceptible

cultivar CO-39 at a rate of $1 \times 10^5$ spores ml$^{-1}$. Images were taken at 5 days post inoculation (dpi). For wounded leaf infection assays, equal sized blocks of mycelia were placed on leaves of 3-week-old rice seedlings of the susceptible cultivar CO-39 that had been abraded with a fine needle. Images were taken at 7 dpi. Live-cell imaging of $\Delta glt1$ occurred at 44 hpi; white arrowheads indicate appressorial penetration sites, stars indicate movement of IH into neighboring cells, scale bars are 10 µm. Representative images and values are derived from observing 50 infected rice cells per leaf sheath per strain. $n = 3$ biological replicates. Values are means ± SD. **d** Plate test images of the indicated strains were taken at 10 days. Gln is minimal media with 1% (w/v) glutamine as the sole carbon and nitrogen source. Glt is minimal media with 1% (w/v) glutamate as the sole carbon and nitrogen source. **e** Rim15 modulates glutaminolysis via (direct or indirect) Gdh2 phosphorylation to promote host colonization. **a, c** Source data (including colony diameter statistics) are provided as a Source Data file.

<10 mM exogenous glutamine did not remediate $\Delta rim15$ IH growth (Supplementary Fig. 5). Thus, sufficient amounts of exogenous glutaminolytic intermediates completely overcome the loss of *RIM15* in *M. oryzae* IH during biotrophic growth in rice, including suppressing host defenses and maintaining biotrophic interfacial membrane integrity.

We next tested whether and how infection by $\Delta glt1$ and $\Delta gdh2$ could be remediated by treatments with exogenous glutaminolytic intermediates. Figure 6a shows that, commensurate with Glt1 being a glutaminolytic enzyme required for glutamine but not glutamate metabolism as a carbon source, treatment with 10 mM glutamate or 10 mM α-ketoglutarate (as DMKG), but not 10 mM glutamine,

remediated $\Delta glt1$ biotrophic colonization of detached rice leaf sheath epidermal cells. Figure 6b shows that on wounded leaf tissue, glutaminolytic intermediates remediated $\Delta rim15$, $\Delta glt1$ and $\Delta gdh2$ infection in a manner that depended on glutaminolytic pathway impairment. Thus, $\Delta rim15$ was remediated by all three intermediates but $\Delta glt1$ was not remediated by 10 mM glutamine and $\Delta gdh2$ was only remediated by 10 mM α-ketoglutarate (as DMKG). These results strongly suggest that exogenous glutaminolytic intermediate treatment requires IH uptake and, in the case of glutamine or glutamate, metabolism to α-ketoglutarate in a manner dependent on fungal glutaminolysis, thus confirming that glutamine is the preferred

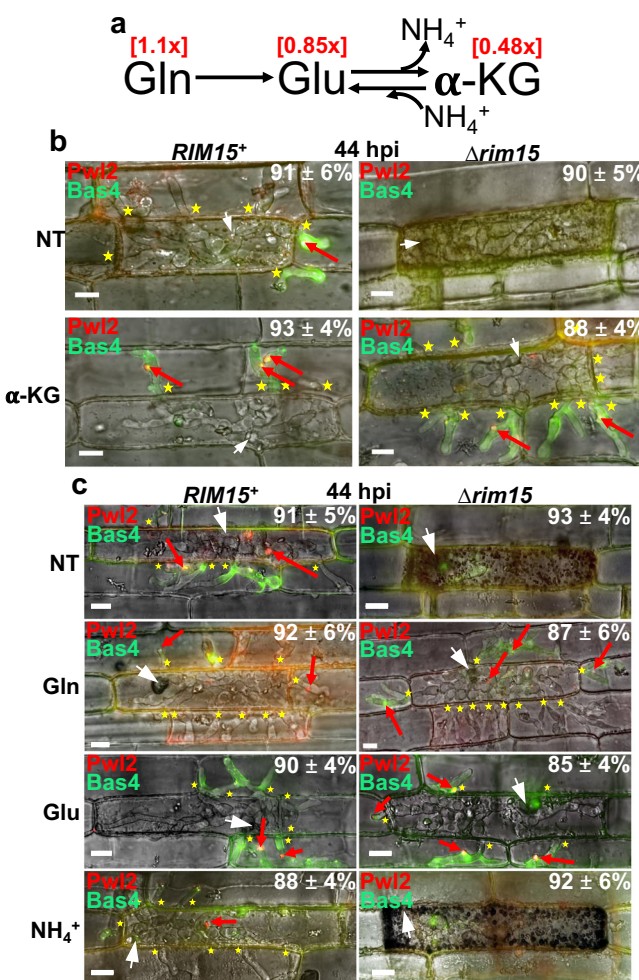

**Fig. 5 | α-ketoglutarate treatment remediates Δ*rim15* biotrophic growth.**
**a** Glutamine and glutamate concentrations were similar in WT and Δ*rim15* mycelia under steady state growth conditions in minimal media with 0.5 mM glutamine as the sole carbon and nitrogen source, but α-ketoglutarate levels were almost halved in Δ*rim15* mycelia compared to WT. Values are the ratios of metabolite concentrations in Δ*rim15* over the metabolite concentrations in WT. Raw values are in Supplementary Data 3. **b** Live-cell imaging of detached rice leaf sheath epidermal cells infected with the indicated strains shows how treatment with 10 mM of the α-ketoglutarate (α-KG) cell-permeable analog dimethyl α-ketoglutarate (DMKG) remediated Δ*rim15* biotrophic growth and restored biotrophic interfacial membrane integrity. **c** Live-cell imaging of detached rice leaf sheath epidermal cells infected with the indicated strains show that treatment at 36 hpi with 10 mM of the glutaminolytic amino acids glutamine and glutamate (as L-Glutamic acid monosodium salt hydrate), but not 5 mM ammonium tartrate (NH₄⁺), remediated Δ*rim15* biotrophic growth in host rice cells by 44 hpi. **b**, **c** White arrowheads indicate appressorial penetration sites. Red arrows indicate BICs. Asterisks indicate movement of IH into neighboring cells. Bars are 10 μm. Merged channel is shown. NT is no treatment. Representative images and values are derived from observing 50 infected rice cells per leaf sheath per treatment. *n* = 3 biological replicates. Values are means ± SD. **b**, **c** Source data are provided as a Source Data file.

carbon source during biotrophic growth. These results also indicate by extension that glutaminolytic metabolites in the host rice cell are insufficient to suppress glutaminolytic defects in *M. oryzae* mutants (otherwise all mutants in Fig. 6b would colonize untreated wounded leaves), suggesting that host-derived glutaminolytic nutrients, which are indeed present in leaf cells in detectable amounts including α-ketoglutarate[28], are not sufficiently acquired by the fungus during biotrophy, possibly in order to prevent detection by the host plant. We conclude that the provision of α-ketoglutarate in the fungal cell

via Rim15-controlled glutaminolysis is an absolute requirement for biotrophic colonization of living host rice cells.

## α-ketoglutarate activates TOR signaling and suppresses autophagic flux

To better understand how α-ketoglutarate promoted Δ*rim15* biotrophic growth, and to determine if Rim15-dependent α-ketoglutarate production was linked to Rim15-controlled autophagy, we assessed the effect of α-ketoglutarate treatment on autophagy induction *in planta*. Compared to the untreated controls, α-ketoglutarate treatment at 36 hpi suppressed autophagy by 44 hpi in both Δ*rim15* and WT IH (Fig. 7a and 7b, respectively), leading to GFP-Atg8 accumulation in the cytoplasm and little GFP accumulation in vacuoles even though both Δ*rim15* and *RIM15*⁺ IH were extensively moving cell-to-cell by this time (Fig. 7a, b). This indicates that autophagy is not required for cell-to-cell spread in the presence of exogenous α-ketoglutarate. Further evidence that autophagy is suppressed by α-ketoglutarate was obtained when *RIM15*⁺ mycelia were switched to water supplemented with 10 mM α-ketoglutarate (as the cell-permeable DMKG analog). Compared to mycelia shaking in water alone (where GFP accumulated in vacuoles) autophagy was suppressed (resulting in GFP-Atg8 accumulation in cytoplasm) in the presence of α-ketoglutarate (Fig. 7c). Thus, α-ketoglutarate treatment suppresses autophagy in the absence of other nutrients including a nitrogen source.

Treatment with α-ketoglutarate suppressed autophagy *in planta* but, in contrast to 3-MA treatment (which suppresses both autophagy and biotrophic growth), α-ketoglutarate treatment stimulated biotrophic growth. To better understand this seeming paradox, we first considered that α-ketoglutarate might act on TOR signaling, which when active promotes growth and suppresses autophagy. To test this hypothesis, we performed Western blot analysis on *RIM15*⁺ mycelia grown in complete media (with and without the TOR inhibitor rapamycin) or after shaking in water (with and without α-ketoglutarate as DMKG) using the anti-phospho-p70 S6 kinase antibody that we have previously shown can assess TOR activity by probing the phosphorylation status of Sch9, the *M. oryzae* functional orthologue of the TOR substrate p70 S6 kinase[14]. Figure 7d shows how, compared to untreated nutrient-rich complete media (CM), Sch9 phosphorylation in mycelia was reduced in CM following treatment with the specific TOR inhibitor rapamycin (as expected) and by shaking in water, which is nutrient-free and therefore TOR inactivating, but Sch9 phosphorylation was not reduced to the same extent when mycelia were exposed to water containing 10 mM α-ketoglutarate. TOR activity status was correlated under these conditions with autophagy activity, whereby Western blot analysis using anti-GFP antibody showed 10 mm α-ketoglutarate treatment in water suppressed GFP-Atg8 processing to free GFP in *RIM15*⁺ mycelia, compared to the control treatments (Fig. 7e). Together, these results indicate that α-ketoglutarate activates TOR signaling to inhibit autophagy activity in the absence of other exogenous nutrients. Furthermore, α-ketoglutarate is a carbon but not a nitrogen source and cannot support fungal growth alone. Thus, we conclude that α-ketoglutarate is a metabolic signal, potentially a carbon and nitrogen sufficiency signal from glutaminolysis, that activates TOR signaling to promote growth.

To further understand why exogenous α-ketoglutarate suppressed autophagy while promoting biotrophic growth, despite autophagy being required for biotrophic growth in untreated IH, we hypothesized that a major role of *RIM15*-controlled autophagy during biotrophy might be to supply α-ketoglutarate for growth. If so, autophagy would be dispensable for biotrophy in the presence of exogenous α-ketoglutarate. Two lines of evidence support this notion. First, we found that α-ketoglutarate treatment could remediate IH growth following autophagy blocking by 3-MA. WT-infected rice cells were treated with 3-MA at 36 hpi. Compared to water controls, replacing 3-MA at 40 hpi with 10 mM α-ketoglutarate (as DMKG) resulted, by 44 hpi, in IH growth, biotrophic interfacial

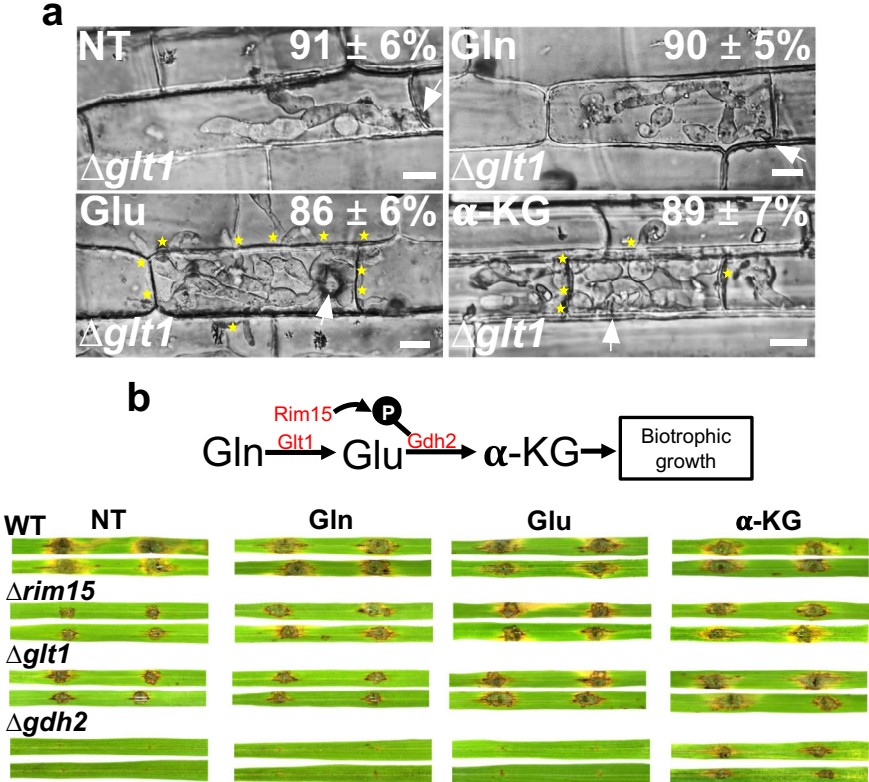

**Fig. 6 | Remediation by exogenous metabolites requires fungal glutaminolysis.**
**a** Live-cell imaging of WT and Δ*glt1* at 44 hpi after treatment with 10 mM of the
indicated metabolites at 36 hpi. α-ketoglutarate was added as DMKG. NT is no
treatment. White arrowheads indicate appressorial penetration sites, stars indicate
movement of IH into neighboring cells, scale bars are 10 μm. Representative images
and values are derived from observing 50 infected rice cells per leaf sheath per
treatment. *n* = 3 biological replicates. Values are means ± SD. Source data are
provided as a Source Data file. **b** Rice leaves from 3-week-old rice seedlings were
detached, abraded with a fine needle, and inoculated with mycelial plugs of the
indicated strains. Following inoculation, wounded sites were treated with 50 μl of
10 mM glutamine, 10 mM glutamate, or 10 mM α-ketoglutarate (α-KG) as DMKG
every 24 h. H₂O-treated leaves were used as the non-treatment control (NT). Leaves
were imaged at 8 dpi.

membrane integrity and tip BICs in new cells (Fig. 7f). Thus, α-
ketoglutarate treatment permits escape from blocked autophagy that
otherwise results in the loss of interfacial membrane integrity and
aborted biotrophy. Second, the *M. oryzae* autophagy mutant Δ*atg8*,
which is unable to form functional appressoria[29], could not colonize
wounded rice leaf tissues from mycelial blocks and cause lesions
unless treated with 10 mM α-ketoglutarate (Fig. 7g). Thus, *ATG8*-
dependent autophagy is essential for rice tissue colonization, but the
loss of *ATG8* can be overcome by α-ketoglutarate treatment. Taken
together, we conclude that autophagy is upstream of α-ketoglutarate,
which is a TOR-activating signaling molecule required for host cell
colonization (Fig. 7h). Furthermore, because autophagy is dis-
pensable in the presence of exogenous α-ketoglutarate, the provision
of α-ketoglutarate via glutaminolysis is a major (perhaps sole) role of
*RIM15*-dependent autophagic cycling during biotrophy.

**Glucose treatment suppresses host plant defenses in Δ*rim15*-
infected rice cells but does not remediate Δ*rim15* biotrophic
growth**
To understand why the provision of α-ketoglutarate is central to the
success of the *M. oryzae* biotrophic growth stage, we next hypothe-
sized that, like in other rapidly proliferating cells[30–33], *M. oryzae*
uses glutamine as a preferred carbon source for growth in order to
preserve glucose for redox homeostasis and scavenging host ROS
(which otherwise triggers plant innate immunity[10]) via its metabolism
through the oxidative pentose phosphate pathway (PPP). In support of
this notion that glucose is important for generating reducing equiva-
lents (ie NADPH) through the PPP, Supplementary Data 4 shows that

oxidized glutathione disulfide was elevated (1.34-fold) in Δ*rim15*
mycelia, and reduced glutathione levels were depleted (0.58-fold),
compared to WT, suggesting impaired antioxidation capabilities in
Δ*rim15*. To test our hypothesis, we treated Δ*rim15*-infected rice cells
with 1% (w/v) glucose at 36 hpi and viewed the cells at 44 hpi. Figure 8a
shows that treatment with glucose suppressed the plant defense
responses observed in untreated Δ*rim15*-infected rice cells but, in
contrast to α-ketoglutarate treatment, did not remediate Δ*rim15* IH
growth. Figure 8b shows that glucose treatment of detached rice leaf
sheath epidermal rice cells infected with the superoxide dismutase
mutant Δ*sod1*, which is unable to detoxify host ROS and thus triggers
plant defenses[34], did not suppress host innate immunity, indicating
exogenous glucose alone is not sufficient to detoxify host ROS and/or
otherwise impair host defense responses. Taken together, we con-
clude that *in planta*, Rim15-dependent autophagy and glutaminolysis
provides α-ketoglutarate to the TCA cycle for growth in order to
conserve glucose for generating reducing equivalents in the PPP that
detoxify host ROS and prevent plant defense activation. In Δ*rim15*, as
α-ketoglutarate levels deplete, glucose might be drawn into the TCA
cycle, impairing antioxidation capacity. This conclusion is consistent
with the observed late onset of plant defense responses in Δ*rim15*-
infected cells (which occurs after at least 15 h of biotrophic growth).

When all our data are considered together, we propose that our
results fit the model in Fig. 9.

## Discussion
Many important eukaryotic filamentous plant pathogens exhibit a
symptomless biotrophic growth stage, where microbial cells grow in

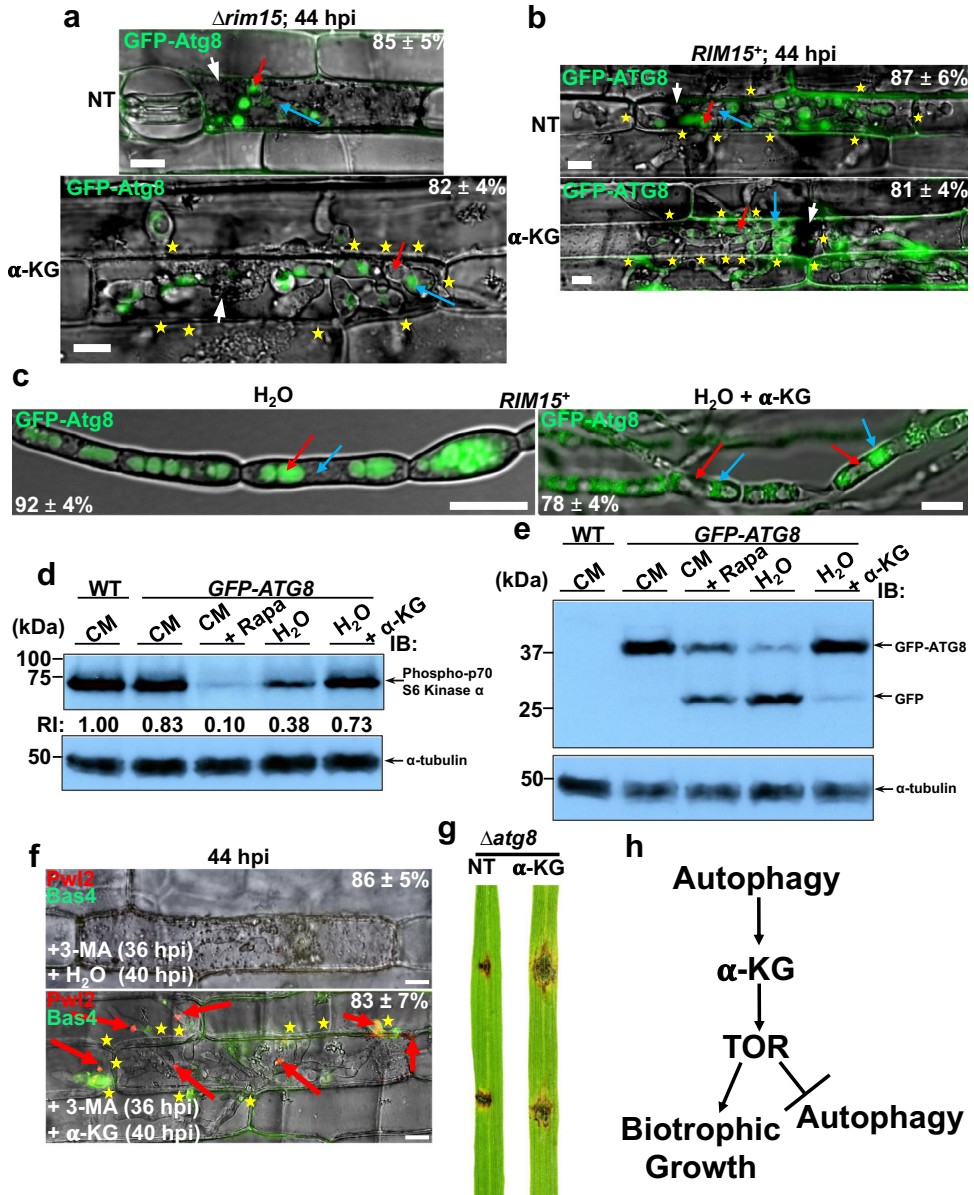

**Fig. 7 | α-ketoglutarate acts downstream of autophagy to activate TOR.** Micrographs showing how 10 mM α-ketoglutarate (α-KG) treatment (as the cell-permable DMKG analog) suppresses autophagy activity in Δ*rim15* (**a**) and WT (**b**) at 44 hpi. DMKG was applied to detached rice leaf sheath epidermal cells at 36 hpi, and images were taken at 44 hpi. **c** Micrograph showing how exposure to water alone for 4 h induced autophagy in WT IH; in contrast, exposure to 10 mM α-ketoglutarate (as DMKG) in water for 4 h suppressed autophagy. **a–c** Examples of vacuoles are indicated with red arrows; examples of cytoplasm are indicated with blue arrows. Bar is 10 μm. Merged channel is shown. **d, e** Western blots showing how, relative to CM media, growth in water suppressed TOR activity as evidenced by decreased Sch9/p70 S6 kinase phosphorylation monitored by the anti-phospho-p70 S6 kinase antibody (**d**) and induced autophagic flux as evidenced by the increase in free GFP compared to GFP-Atg8 monitored by the anti-GFP antibody (**e**). However, TOR activity was increased (**d**) and autophagic flux was decreased (**e**) in water following treatment with 10 mM α-ketoglutarate (α-KG) as DMKG. RI = relative intensity calculated by normalizing Sch9 phosphorylation levels determined

using the anti-p-p70 S6 kinase antibody against tubulin α levels determined by the anti-tubulin α antibody. Rapa is 1 μM rapamycin. The experiments were performed three times with similar results. **f** Live cell imaging of detached rice leaf sheath epidermal cells infected with WT and treated as indicated. 3-MA is 10 mM 3-methyladenine. α-KG is 10 mM α-ketoglutarate as DMKG. Bar is 10 μm. Merged channel is shown. **a, b, c, f** Representative images and values are derived from observing 50 infected rice cells per leaf sheath per treatment (**a, b, f**), or from 50 vegetative hyphae (**c**). *n* = 3 biological replicates. Values are means ± SD. **a, b, f** Asterisks indicate movement of IH into neighboring cells. Red arrows indicate BICs. White arrowheads indicate appressorial penetration sites. NT is no treatment. **g** Wounded rice leaves inoculated with mycelial plugs of the Δ*atg8* strain treated with 50 μl of 10 mM α-ketoglutarate (α-KG) as DMKG and imaged at 8 dpi. NT is no treatment. **h** Schematic showing relationships between TOR, autophagy, α-ketoglutarate (α-KG) and biotrophy. **a–f** Source data are provided as a Source Data file.

living host plant cells separated from host cytoplasm by extensive biotrophic interfaces[35], but little is known about the molecular and metabolic drivers involved, limiting the discovery of novel intervention strategies. Here, following an initial period of Rim15-independent biotrophic growth requiring active TOR signaling[36], we showed how

Rim15 coordinates cycles of autophagy with glutaminolysis (the latter via phosphorylation control of Gdh2) to provide α-ketoglutarate, both as a TOR reactivation signal and as an anaplerotic substrate for the TCA cycle. The preferred use of glutamine via α-ketoglutarate as a carbon source for IH growth protects glucose from consumption

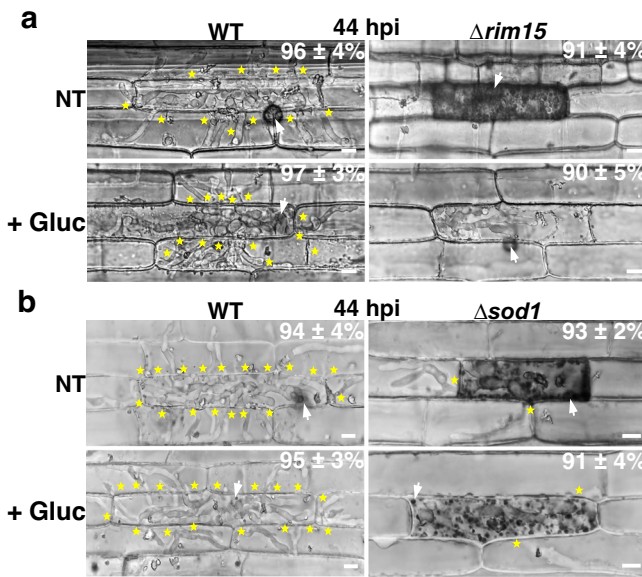

**Fig. 8 | Glucose treatment suppresses host defenses but does not remediate biotrophic growth in Δ*rim15*-infected rice cells. a, b** Live-cell imaging at 44 hpi of the indicated strains after treatment with 1% (w/v) glucose (+Gluc) at 36 hpi. NT is no treatment. White arrowheads indicate appressorial penetration sites, stars indicate movement of IH into neighboring cells, scale bars are 10 μm. Representative images and values are derived from observing 50 infected rice cells per leaf sheath per treatment. *n* = 3 biological replicates. Values are means ± SD. Source data are provided as a Source Data file.

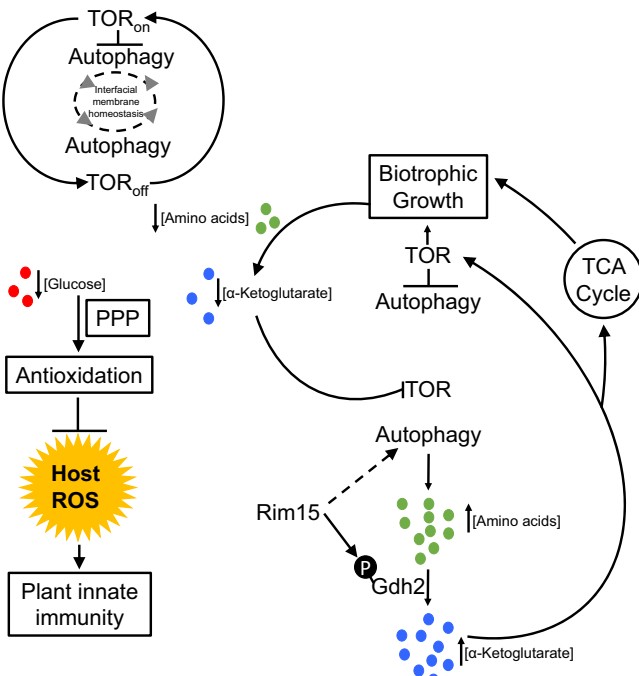

**Fig. 9 | Model for the metabolic control of biotrophy.** Considering the autophagic cycling observed in WT during biotrophy, we propose that, following the Rim15-independent, TOR signaling-dependent establishment of early biotrophic growth[36], Rim15 coordinates autophagy with glutaminolysis under nutrient depleted conditions to generate α-ketoglutarate. α-ketoglutarate drives biotrophy by acting both as a nutrient sufficiency signal to reactivate TOR, and as a TCA cycle-replenishing metabolite. This strategy preserves limiting amounts of glucose for metabolism through the oxidative pentose phosphate pathway (PPP), thereby generating reducing equivalents (NADPH) that activate antioxidants to maintain redox homeostasis and detoxify host ROS. As α-ketoglutarate levels fall during growth, TOR signaling is inactivated, and the cycle starts again. Concomitantly, autophagy maintains membrane homeostasis and biotrophic interfacial membrane integrity under α-ketoglutarate-depleted conditions.

for fuel and instead conserves it for generating reducing equivalents (NADPH) through the PPP that activate the antioxidants needed to scavenge host ROS. NADPH is itself a signal for carbon metabolism and antioxidation gene expression[21,37,38]. The provision of α-ketoglutarate is the major role of autophagy during biotrophy, and autophagy and glutaminolysis are dispensable in the presence of exogenous α-ketoglutarate, whereas glucose treatment only suppresses host defenses.

The TOR-reactivating role deduced here for α-ketoglutarate is in line with studies showing how α-ketoglutarate production from glutaminolysis, and treatment with the cell-permeable α-ketoglutarate analog DMKG, stimulated mTORC1 activation in human cells[39]. Moreover, supplementation with glutamine alone was sufficient to restore mTORC1 activity in mouse embryonic fibroblasts (MEFs) following a prolonged period of amino acid starvation, and mTORC1 reactivation after starvation in MEFs was autophagy-dependent[40]. Thus, our findings have implications for understanding cellular behavior across eukaryotic systems. Finding evidence that autophagy supplies α-ketoglutarate during biotrophy is also significant because although the function of autophagy in cells is well understood, few physiological roles—including how autophagy-derived metabolites are used by cells—are known[41].

We conclude that Rim15 controls *M. oryzae* metabolism during biotrophy by coordinating autophagy with glutaminolysis in order to supply the fungal cell-intrinsic metabolite α-ketoglutarate for TOR reactivation and to fuel growth while preserving glucose for antioxidation and host plant defense suppression. This work provides a solid foundation for future studies probing and interpreting fungal metabolism in host cells. Such studies might foster the discovery of new and targetable pathogen weaknesses with broad applicability.

## Methods
### Fungal strains and growth conditions
The *M. oryzae* Guy11 rice isolate was used as the wild type (WT) strain for this study[2]. Strains used in this study are listed in Supplementary

Data 5 and are maintained as filter stocks in the Wilson lab. Strains were grown at 26 °C on solid or in liquid complete media (CM) or minimal media[14,42]. Plates were incubated at 26 °C under 12 h light/dark cycles for 10–15 days. Plate images were taken with a Sony Cyber-shot digital camera, 14.1 megapixels. For vegetative hyphal treatments, strains were grown in shaking CM media for 42 h and then washed with distilled water three times. Washed mycelia were transferred to the indicated treatments and incubated at 26 °C for up to 4 h, as stated in the figure legends.

### Generation of targeted gene deletion mutant strains by homologous recombination
Genes were disrupted in *M. oryzae* using the split marker technique[21] and the primers listed in Supplementary Data 6. *RIM15*, *GDH1* and *GDH2* coding regions were replaced with the *ILV2* gene conferring resistance to sulphonyl urea. The *GLT1* coding region was replaced with the *hph* gene conferring hygromycin resistance. Fungal protoplasts were transformed with the respective DNA fragments[42], and transformants were selected for sulphonyl urea or hygromycin resistance and confirmed by PCR. At least two independent deletants per gene knockout were characterized fully and shown to be indistinguishable. Examples of gene knockout confirmation by PCR are shown in Supplementary Fig. 6.

### Vector construction and transformation
*M. oryzae* gene sequences obtained from the *M. oryzae* genome database (http://fungi.ensembl.org/Magnaporthe_oryzae/Info/Index) were

used to design the primers in Supplementary Data 6 that were then used for constructing the vectors in Supplementary Data 7. To express *GFP-ATG8*, the full-length coding region of *ATG8* was amplified from Guy11 genomic DNA using the RP27-NGFP-ATG8-F/-R primer pairs. The end of the PCR product contained 15–20 bases matching the ends of the NotI-linearized pGTN vector. The purified PCR product was fused with linearized pGTN, using the In-Fusion HD enzyme kit, placing it in-frame with *GFP* and downstream of the RP27 constitutive promoter. The reaction mixture was transferred into the *E. coli* DH5a strain with ampicillin antibiotic screening, and all colonies were identified by PCR and plasmid integrity was verified by sequencing. The vector was then transformed into WT and Δ*rim15* protoplasts, respectively. All *M. oryzae* transformants were screened for geneticin resistance and confirmed on plants. Five independent transformants per parental strain were characterized and found to have identical growth and infection phenotypes. A similar procedure was used to construct the *RIM15-GFP* complementation vector, except pGTN was linearized with HindIII and BamHI to insert *RIM15* upstream of *GFP*. To generate Δ*rim15* strains producing Pwl2-mCherry:NLS and Bas4-GFP, the Δ*rim15* parental strain was transformed with the pBV591 vector[5] and selected using hygromycin resistance. Three independent transformants were fully characterized and found to have identical phenotypes. The *Δglt1* mutant strain was complemented with *GLT1* using the plasmid *GTL1 pFL6*, whereby *GLT1* was subcloned into pFL6 using the yeast gap repair approach[42], and transformants were selected for neomycin resistance. The Δ*gdh2* deletant was complemented with *GDH2* or *GDH2^{S24A}* (encoding the phospho-null version of Gdh2), and Δ*rim15* was complemented with *GDH2^{S24D}* (encoding the phospho-mimicking version of Gdh2), by transformation with pGTN vectors carrying the respective *GDH2* coding sequences synthesized by Synbio Technologies (USA). At least three transformants carrying *GDH2*, *GDH2^{S24A}* or *GDH2^{S24D}* sequences were selected by hygromycin resistance and confirmed on plant hosts.

## Plant inoculations and live-cell imaging

For whole plant inoculations, spores of the indicated strains were harvested from colonies grown on solid oatmeal agar plates for 14 days and applied to 3-week-old rice seedlings of the susceptible cultivar CO-39 at a rate of $1 \times 10^5$ spores ml$^{-1}$. Images were recorded at 5 days post inoculation. For live-cell imaging, spores of the indicated strains were harvested from 14-day-old colonies grown on solid oatmeal agar plates and inoculated at a rate of $2 \times 10^4$ spores ml$^{-1}$ onto detached rice leaf sheaths from 4-week-old rice seedlings of the susceptible cultivar CO-39[14]. At least three leaf sheaths from three different plants were used per experiment. Leaf sheath epidermal cells were imaged at the indicated times using a Nikon Eclipse Ni-E upright microscope and NIS Elements software. Unless otherwise specified, treatments were added at 36 hpi and viewed at 44 hpi. Treatments were added at the concentrations indicated. Treatments were purchased from Sigma-Aldrich, USA. Infected rice cells were stained with 1 mg ml$^{-1}$ 3,3′-diaminobenzidine, and spores were treated with 0.4 μM of the NADPH oxidase inhibitor DPI[10]. Rice cells were treated with 10 μM rapamycin (Rapa)[14]. All treatment assays were performed at least in triplicate and representative images are shown. For Trypan Blue staining, detached rice leaf sheaths were inoculated at a rate of $0.4 \times 10^5$ spores ml$^{-1}$ with the indicated strains. Leaf sheaths were sliced at 39 hpi and 42 hpi, immersed in a fresh 0.4% solution of Trypan Blue (Millipore Sigma, catalog #93595) and incubated in the dark with gentle shaking for 50 min at room temperature. After six washes with dH$_2$O for 10 min each, the destained leaf sheaths were imaged using a Nikon Eclipse Ni-E upright microscope. For ethanol treatment, sliced rice leaf sheaths were treated with 30% ethanol and then incubated at 24 °C in the dark for 1 h; meanwhile, leaf sheaths treated with dH$_2$O were used as the non-treatment (NT) control. After ethanol treatment, the leaf sheaths were washed extensively with dH$_2$O and then inoculated with spore

suspensions at a rate of $0.4 \times 10^5$ spores ml$^{-1}$ for each strain. The inoculated leaf sheath epidermal cells were imaged at 42 hpi using a Nikon Eclipse Ni-E upright microscope.

Detached rice leaves were wounded by abrasion using an inoculation needle. Mycelial plugs of the indicated strains were placed on the damaged area and incubated for 5 days. Images were taken using an Epson Perfection V550 scanner.

Detached rice leaf sheath assays and artificial hydrophobic surfaces were used to quantify appressorium formation rates. Appressorium penetration rates and rates of IH cell-to-cell movement to adjacent cells were determined using detached rice leaf sheaths. Appressorial formation rates were determined by examining how many of 50 spores had formed appressoria by 24 hpi on artificial hydrophobic surfaces (after applying at a rate of $1 \times 10^4$ spores ml$^{-1}$) or on detached rice leaf sheath surfaces, repeated in triplicate. Penetration rates were determined by examining how many of 50 appressoria on one detached rice leaf sheath surface had penetrated into underlying epidermal cells by 30 hpi, repeated in triplicate. Rates of cell-to-cell movement were determined by observing how many of 50 primary infected rice leaf sheath epidermal cells had formed IH in adjacent cells by 48 hpi, repeated in triplicate.

## Protein extraction, proteomics and phosphoprotein enrichment

Strains were grown as mycelia in liquid CM for 42 h before shifting to minimal media with 0.5 mM glutamine as the sole carbon and nitrogen source for 16 h.

For the proteomics experiments, 0.3 g of wet mycelia per strain (in triplicate) was lysed in 1 mL lysis buffer consisting of 7 M urea, 2 M thiourea, 5 mM DTT, 100 mM tris/HCl pH 7.8, and containing 1X complete EDTA-free protease inhibitor and 1X PhosStop phosphatase inhibitor, for 10 min at 20 Hz on a mechanical tissue lyser. Protein was precipitated with acetone and the pellet was washed and redissolved in the lysis buffer containing 2X PhosStop phosphatase inhibitor. The proteins were assayed using the CBX kit (G-Bioscience). An aliquot of proteins was reduced with DTT and alkylated with iodoacetamide prior to digestion with LysC and then trypsin. Each digest was analyzed by nanoLC-MS/MS using a 2 h gradient on a Waters CSH 0.075 mm × 250 mm C18 column feeding into an Orbitrap Eclipse mass spectrometer. However, an unknown component interfered with the chromatography and samples were rerun after offline solid-phase C18 clean-up (Waters SepPak, 100 mg 1 cc syringe cartridges).

For phosphoprotein enrichment, of the 1 mg of wet mycelia from each strain (in triplicate), 0.95 mg was used for TiO2 phosphopeptide enrichment with lactic acid. Samples were subjected to offline solid-phase C18 clean-up (Waters SepPak, 100 mg 1cc syringe cartridges). Each cleaned sample was then analyzed by nanoLC-MS/MS using a 2 h gradient on a Waters CSH 0.075 mm × 250 mm C18 column feeding into an Orbitrap Eclipse mass spectrometer.

Quantification of the proteins and phosphoproteins was performed separately using Proteome Discoverer (Thermo; version 2.4). All MS/MS samples were searched using Mascot (Matrix Science, London, UK; version 2.6.2). Mascot was set up to search the cRAP_20150130.fasta (124 entries); uniprot- refprot_UP000009058_Magnaporthe_oryzae 20210511 (12,791 sequences); assuming the digestion enzyme trypsin. Mascot was searched with a fragment ion mass tolerance of 0.06 Da and a parent ion tolerance of 10.0 PPM. For the proteomics experiment, deamidation of asparagine and glutamine and oxidation of methionine were specified in Mascot as variable modifications while carbamidomethyl of cysteine was specified as fixed modification. For the phosphoproteomics experiment, deamidation of asparagine and glutamine, oxidation of methionine, phosphorylation of serine, threonine and tyrosine, and acetylation of N-term were specified in Mascot as variable modifications while carbamidomethyl of cysteine was specified as fixed modification. Peptides were validated by Percolator with a 0.01 posterior error probability threshold. The data were searched using a decoy

database to set the false discovery rate to 1% (high confidence). Only proteins with a minimum of two peptides and five PSMs were reported. For phosphoproteins, the minimum was one phosphopeptide and three PSMs. The localization probability of the phosphorylation sites was calculated using PtmRS[43]. Probabilities are indicated in parenthesis next to the amino acid residue. If there is no probability indicated, this means that the phosphorylation of the peptide was not confidently localized.

The peptides and phosphopeptides were quantified using the precursor abundance based on intensity. Normalized and scaled abundances are reported. The peak abundance was normalized using total peptide amount. The peptide group abundances are summed for each sample and the maximum sum for all files is determined. The normalization factor used is the factor of the sum of the sample and the maximum sum in all files. Then, abundances were scaled so that the average of the abundances is equal to 100 for each sample. The imputation mode was used to fill the gaps from missing values. The protein and phosphoprotein ratios were calculated using summed abundance for each replicate separately and the geometric median of the resulting ratios is used as the protein ratios. The significance of differential expression is tested using a $t$ test which provides a $P$ value and an adjusted $P$ value using the Benjamini-Hochberg method for all the calculated ratios.

### Metabolite extraction and metabolomics

Strains were grown as mycelia in liquid CM for 42 h before shifting to minimal media with 0.5 mM glutamine as the sole carbon and nitrogen source for 16 h. Samples were lyophilized before metabolite extraction.

For the metabolomic results in Supplementary Data 3, an aliquot of 20–25 mg of each sample was extracted for polar compounds using the chloroform/methanol/water extraction (Folch Method). The extracts were dried down and derivatized for GCMS using MSTFA + TMCS. The samples were run alongside the mixture of retention index C10-25 alkanes for identification. The data was analyzed using MS-Dial (https://doi.org/10.1038/nmeth.3393) for peak detection, deconvolution, alignment, quantification and identification. The library used was the curated Kovats RI (28,220 compounds, which includes the Fiehn, RIKEN and MoNA databases). The peaks were reviewed and the final list of compounds with RI similarities >90% were reported. Highlighted in red in Supplementary Data 3 are compounds with ID confirmed by authentic standards or by complementary search of the NIST14 library using NIST-MS. The peak area was normalized based on the total ion chromatogram for the assigned metabolites using sum normalization in MetaboAnalyst 5.0.

For the metabolomic results in Supplementary Data 4, the samples were suspended in 500 μL of extraction solution on dry ice, and disrupted by four cycles of the bullet blender at a setting of "7" for 3 min each. The suspension was centrifuged for 10 min at 15,000 × $g$. The supernatant was transferred into a second tube and evaporated using a speed Vac at 4 °C. The pellet was resuspended into 100 μL of LC-Grade water and transferred into V-vials and placed in the autosampler. The sample concentrations are reported in micromolar units, and the metabolite levels were normalized by weight.

### Western blot analysis

For S6K phospho-status analysis, indicated strains were inoculated in liquid CM medium and shaking cultured for 42 h in a 25 °C shaking incubator. After being excessively washed with $H_2O$, the vegetative hyphae were aliquoted into fresh liquid CM, CM containing 1 μM rapamycin, $H_2O$, and $H_2O$ containing 10 mM DMKG (Dimethyl 2-oxoglutarate, a permeable version of α-ketoglutarate), respectively. Following 4 h of further shaking culture, the vegetative hyphae were harvested and frozen in liquid nitrogen and then lyophilized using a Labconco Benchtop Freezone Freeze-Dryer (Freezone 2.5). For immunoblotting analysis, equal amounts of pulverized mycelia were resuspended in denaturing lysis buffer (62.5 mM Tris HCl, pH 6.8, 3% SDS, 10% glycerol,

5% 2-mercaptoethanol) supplemented with protease inhibitors (200 mM AEBSF, 20 mM Bestatin, 5 mM E-64, 10 mM Leupeptin, 10 mM Pepstatin A, 500 mM 1,10-Phenanthroline, 5 mM EDTA, 1 mM PMSF) and phosphatase inhibitors (20 mM NaF, 0.2 M okadaic acid, 20 mM b-gly-cerophosphate, 5 mM Na3VO4), followed by denaturation at 95 °C for 5 min. After clearing the cell lysates by centrifugation at 16,000 $g$ for 15 min at 4 °C, equal volume of the lysate per sample were resolved in a 12% (wt/vol, polyacrylamide) SDS-PAGE gel and blotted onto an Immun-Blot PVDF membrane (Bio-Rad, USA). For phospho-p70 S6 kinase detection, the membrane was first incubated in the blocking buffer (5% BSA in 1xTBS) for 12 h at 4 °C, and then incubated with anti-phospho-p70 S6 kinase α antibody (monoclonal, produced in mouse, Santa Cruz Biotechnology, USA; 1:1000 dilution) in 1xTBST containing 5% BSA for 16 h at 4 °C. After being washed in 1xTBST for three times with 5 min each, the membrane was incubated with anti-mouse IgG-peroxidase (produced in goat, Sigma, USA; 1:10,000 dilution) in 1xTBST containing 5% BSA for 12 h at 4 °C. After being washed in 1xTBST for three times with 5 min each, the blot was imaged using Clarity Western ECL chemiluminescent system (Bio-Rad). Following stripping the anti-phospho-p70 S6 kinase α antibody off the membrane, as a loading reference, α-tubulin was visualized using anti-Tub(α) antibody (monoclonal, produced in rat, Santa Cruz Biotechnology, USA; 1:1000 dilution) in 1xTBS containing 5% non-fat milk for 1 h at room temperature, followed by probing with anti-rat IgG-peroxidase (produced in goat, Santa Cruz Biotechnology, USA; 1:10,000 dilution) in 1xTBST containing 5% non-fat milk for 1 h at room temperature. The imaged blots were quantitated by densitometry using ImageJ analysis software (imagej.net/Welcome). Relative signal intensity of the phospho-p70 S6 Kinase was obtained by normalizing against α-tubulin and then by correcting for the background determined from a WT control strain. To eliminate protein degradation and protein dephosphorylation, low temperature (4 °C), protease inhibitors and phosphatase inhibitors were applied throughout the western blot analysis, including gel running, protein transfer, membrane blocking, and antibody binding, and membrane washing steps.

To monitor autophagy in each sample, the intact GFP-Atg8 and the cleaved free GFP were visualized using Anti-Green Fluorescent Protein (GFP) antibody (monoclonal, produced in mouse, Sigma, USA; 1:1000 dilution) in 1xTBS containing 5% non-fat milk for 2 h at room temperature, followed by probing with anti-mouse IgG-peroxidase (produced in goat, Sigma, USA; 1:10,000 dilution) in 1xTBST containing 5% non-fat milk for 1 h at room temperature. As a loading reference, α-tubulin in all samples was visualized using anti-Tub(α) antibody following stripping the GFP antibody off the membrane.

### Statistical analysis of fungal growth and infection-related development

Microsoft Excel 2016 was used to calculate the means ± SD for the fluorescent protein distribution patterns, vegetative hyphal growth and invasive hyphal growth shown in Fig. 1c–e; 2a–d; 3h,i; 4a,c; 5b,c; 6a; 7a–c,f; 8a,b; and Supplementary Figs. 1f, 3, and 5. Mean values for sporulation rates, appressorium formation rates, appressorial penetration rates and invasive hyphae cell-to-cell movement rates (Supplementary Fig. 1b,d) were compared using the one-way analysis of variance function in PASW Statistics 18.0 (PASW Statistics Inc.) with the Tukey HSD multiple comparison test.

### Reporting summary

Further information on research design is available in the Nature Portfolio Reporting Summary linked to this article.

## Data availability

All data supporting the findings of this study are available in the manuscript or in Supplementary Information. The *M. oryzae RIM15*, *GLT1*, *GDH1*, *GDH2*, *ATG8*, *SOD1*, *PWL2*, and *BAS4* gene sequences are available at NCBI under the accession numbers MGG_00345

(https://www.ncbi.nlm.nih.gov/gene/2675109), MGG_07187, MGG_08074, MGG_05247, MGG_01062, MGG_02625, MGG_13863 and MGG_10914, respectively. Uniprot accession numbers can be accessed at https://www.uniprot.org. The mass spectrometry proteomics data generated in this study have been deposited to the ProteomeXchange Consortium via the PRIDE partner repository (http://www.ebi.ac.uk/pride) with the dataset identifier PXD04307. Metabolomic datasets are available upon request. Uncropped western blots are provided in Supplementary Information. Mutant strains generated in this study are available from the corresponding author with an appropriate APHIS permit. Source data are provided with this paper.

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

## Acknowledgements

This work was supported by National Science Foundation funding (IOS-1758805 and IOS-2106153) to R.A.W. Z.G. was supported by funding from the China Scholarship Council. We thank Dr. Sophie Alvarez of the Center for Biotechnology, University of Nebraska-Lincoln, for the (phospho)proteomic and metabolomic analyses in Supplementary Data 1–3. We thank Dr. Javier Seravalli of the Redox Biology Center, University of Nebraska-Lincoln, for the metabolomic analysis in Supplementary Data 4.

## Author contributions

R.A.W. conceived the project and obtained funding. G.L. and R.A.W. designed the experiments and interpreted the data. G.L., Z.G., N.D., M.M.-G., R.O.R., and M.R. performed the experiments. R.A.W. wrote the manuscript and composed the figures, with contributions from all authors.

## Competing interests

The authors declare no competing interests.
