## [Peer Review File · Nature Communications]

A protein kinase coordinates cycles of autophagy and glutaminolysis in invasive hyphae of the fungus *Magnaporthe oryzae* within rice cellsREVIEWER COMMENTS

Reviewer #1 (Remarks to the Author):

The manuscript submitted by Li et al. report on the association between TOR signalling pathways with fungal invasive growth in *M. oryzae*. The study produced a series of nice imaging data to demonstrated RIM15, a gene which affects integrity of EHIM, and its association with α -ketoglutarate. Combining with further analysis, the authors demonstrate Rim15-dependent cycles of autophagic flux liberate α -ketoglutarate as an amino acid-sufficiency signal to trigger TOR reactivation and promote biotrophic growth in host rice cells. Overall, I find this study is of great novelty, however, there are still a number of potential problems with current reported evidence and conclusions that need to be addressed.

Major concerns:

1. According to the results sections written in the manuscripts (line 77-141), the most significant phenotype of compromised Rim15 in *M. oryzae*, is its abolished colonization after biotrophic growth. The evidences of imaging of Bas4-GFP, and PWL2-RFP is a good indication to elucidate EHIM and exam the integrity of EHIM. However, it is ideal and rigorous to use plant transgenic marker line, given that the EHIM is plant-membrane derived structure. The observation of Bas4-GFP, and PWL2-RFP in leaked host cell, is an indirect way to determine integrity of EHIM, only means the secretory pathways are affected, but not directly indicate a defect in biotrophic growth. Considering the plant transgenic marker line is not easy to obtain, there could be also possible to stain the IH with Trypan blue, which is usually used to quantify live cells by labeling dead cells exclusively. Because live cells have an intact cell membrane, trypan blue cannot penetrate the cell membrane of live cells and enter the cytoplasm. In a dead cell, trypan blue passes through the porous cell membrane and enters the cytoplasm.
2. Another point I would like to rise, is that the leakage first invaded cell is not necessarily means its plant defence response or impaired/abnormal invasive growth from pathogen side, especially for the case of Rim15. Therefore, the statement of "Rim15 acted in the TOR-Imp1-autophagy signalling cascade downstream of TOR but upstream of Imp1 at a branch with a separate Rim15-controlled pathway mediating ROS scavenging and host innate immunity suppress" is clearly not convincing. Although, the ROS assay experiment was performed, but it is not sufficiently strong to support the assumption about "host innate immunity suppress". A good practice is using ethanol-killed leaf sheath to perform the invasive growth assay, and see if the mutant could continue its further growth as wild type, or trapped in the first cell. Therefore, I really wonder if authors could re-consider the whole assumptions made in the manuscript, as the imaging observation in this section is the foundation of the whole study.
3. The authors concluded that autophagic cycling in IH is Rim15-dependent, this is questionable. This conclusion just deduced from the result that the autophagic cycling is abolished in Δ Rim15. The defect in autophagic cycling of Δ Rim15 could also possibly be a consequence of host defense-associated inhibition, not absolutely means Rim15 is involved in autophagic cycling. Therefore, I think it is not a solid conclusion.
4. Line 115-116, "However, about 4 % of Δ Rim15-infected rice cells displayed some Bas4-GFP leakage into the host cytoplasm." (Fig. 1d). How did the authors know that this is a leakage? Could that be a plant side defense response? Because autofluorescence could be found in some incompatibility situation during *M. oryzae* infection (as we have observed). It's lack of evidence to confirm that this situation is a Bas4-GFP leakage into the host cytoplasm.
5. I think the multi-omics bioinformatics analyses are really too simple, some important information should be provided. There is an important step of RNA-seq as batch effects removal should be executed before DEGs analysis. Sequence depth of leaf sheath of RNA-seq should be provide, as the ratio of plant/pathogen would be significantly high. It is essential to assess the quality of the data carefully, given that the result of PCA is not completely repeated between the replicates. How many genes of *M. oryzae* are covered from all the samples? Also, the multi-omics data has not been submitted to database yet.

6. It is required to detect the level of α -Ketoglutarate (α -KG) in planta. For example, it is possible to perform an α -Ketoglutarate (α -KG) fluorometric detection assay, which provides a fluorescence-based method for quantifying α -KG.

Minors:

1. It would be helpful and great to take the message if the images in the figures could be arranged in a decent way. For instance, Figure 4, Figure 5, Figure 7, Fig S2, Fig S3.
2. Fig. 1b, the lesions are not typical.
3. Fig. 3a, needs statistical data.
4. Fig. 3b, scale bar should be in similar place for consistence.
5. Fig. 3b, needs statistical data.
6. Fig. 6g, needs statistical data.
7. Fig. 10a, needs statistical data.

Reviewer #2 (Remarks to the Author):

This study has primarily found a critical role for Rim15 in regulating biotrophic growth in *M. oryzae*. I will first summarize my main take on this paper: the central findings are novel and extremely exciting, and should be published. I also think this manuscript suffers from too much data (which not just takes away from central points, but creates multiple new tangents). The manuscript suffers terribly because (i) for a large part, the statements, inferences (and title) are distinct from the main findings, (ii) at many parts of the manuscripts, an inference (or conclusive statement) precedes the actual data shown, and (iii) a lot of data is included that distracts (or actually diverts away from) the main findings. My review below will have two parts, broader comments, and a few very specific comments.

General comments:

1) the title suggests a role for autophagy-dependent TOR reactivation in fungal growth in living host rice cells. While the data presented is excellent, it has not been presented in the form of a coherent story that supports the title. The paper requires rewriting and changing of title and figure panels to support one coherent story. For example, a title of 'Rim15/greatwall kinase drives fungal growth in living host rice cells, controlling autophagic flux' (or something vaguely like that) would be far more appropriate. The data presented currently, and the title are quite different.

2) The authors show a role of Rim15 in regulating biotrophic growth, specially post 39hpi. They then show that this might be because Rim15 regulates autophagy (in rim15 cells, activation of autophagy at 36hpi by AM or Rapa rescues the rim15 phenotype at 44hpi). rim15 cells have a delay in autophagy activation, which suggests Rim15 function is important for temporal activation of autophagy during biotrophic growth. Further, autophagic flux cycles during this growth, which does not happen in a rim15 mutant. However, there are some large gaps in the final conclusions. First, the argument that autophagy cycling is important for biotrophic growth has not been demonstrated (but should be implied), as inhibition of autophagy at any time point leads to a defect in biotrophic growth. Further, even though autophagy might be functional post 40hpi in rim15 cells, biotrophic growth is lost, so at what time during biotrophic growth is Rim15 dependent activation of autophagy required?

3) rim15 mutants have a differential transcriptional profile with multiple metabolic nodes affected. This in itself is not surprising. However, the authors then focus on one of these nodes - glutamine metabolism. Here, the first argument that GO-GOGAT directly regulated is not convincing at all (based on just gene expression). The authors should hence clarify that the difference is at protein levels. Glt1 (GOGAT) transcripts hardly change, but protein decreases (which implies very different types of regulation). Next, the authors provide data that supports a role for glutamine metabolism, but not really GS-GOGAT. Is GOGAT itself central to their story? Not really, but the presentation makes the case that it is. Metabolic profiling shows a decrease in steady-state Glu and α -KG levels in rim15 cells. Further, rim15 mutants are not able to utilize glutamine effectively irrespective of

presence or absence of glucose. The authors should clarify that the growth defect in rim15 is also seen in presence of glucose.

4) Hence, the argument is that rim15 phenotype of decreased biotrophic growth is due to decreased a-KG in rim15 cells, which is consistent with an a-KG growth rescue. However, here the authors suggest that this is due to a rescue of autophagy suppression at 44hpi (by adding a-KG at 36hpi). However, at 44hpi autophagy is highest during biotrophic growth (Fig. 5C). So, the logic of why the suppression of autophagy would rescue the phenotype is entirely unclear. How do the authors explain this? A reinterpretation of the data would be warranted. Along these lines, does a-KG addition at 36hpi affect biotrophic growth at 44hpi in RIM15+ cells since autophagy is affected?

5) Even though S6Kinase phosphorylation increases post addition of a-KG in water, it is not clear how this is related to/can be contextualized w.r.t biotrophic growth. What appears to be the case is that a-KG has a metabolic role, rather than signalling role. Further, a-KG addition activating TOR should result in less autophagy and directly contradicts the starting model of autophagy activation at 36hpi (and even TOR inhibition by Rapa) being required for biotrophic growth at 44hpi, in context of rim15 mutants. How do the authors clarify these opposing data? And again, the title of the paper is not justified from the data. Going by their data, the role of autophagy dependent TOR activation is not the focus of the paper (and Rim15's role certainly is).

6) Relatedly, glutamine metabolism is important for regulating biotrophic growth. Consistent with this, Gln/Glu addition in rim15 mutants also rescue the phenotype. All these data does not differentiate between metabolic or signalling roles for a-KG being important. The authors have overemphasized a signalling role for a-KG. Further, a-KG levels will ideally be high in GS or GOGAT inhibited cells, as a-KG is either a direct or an indirect substrate for both enzymes, So why will inhibition of GS/GOGAT lead to a rim15 like phenotype?

7) Figure 9 is a complete diversion from the main topic of 'how might Rim15 be playing a role in regulating biotrophic growth'. IT is not clear how these data fit into this story at all. If the choose to keep it here, the authors must explain why Asn addition in asn1 mutant rescues biotrophic growth, but does not do so in rim15 cells, and why in these cells activation of autophagy by AM not rescuing the phenotype? This appears to be a major (and unconnected) diversion of this study.

8) The final data suggest that even during autophagy inhibition, addition of a-KG rescues biotrophic growth in WT cells. This argues that a-KG (or an immediate metabolic product of a-KG, glutamate or succinate) is the reason why autophagy is required. The paper ends with a circular argument of a-KG being needed at 36hpi for autophagy inhibition which rescues rim15 phenotype, but simultaneously autophagy activation at 36hpi is important for a-KG production. Further, as mentioned previously, in WT cells, autophagy is highest at 44hpi, not inhibited. How can the authors explain this incongruity?

Some more specific comments:

1) Figure panels should be changed, so that all data supporting one argument can all be included in one single figure. The data supporting the same points seem to be divided between figures.

2) All throughout papers, panels having separate data should be called out separately, and must state what they say. E.g. Supplementary fig 1D and figure 6.

3) Lines 64-66: the data for these proteins being important for maintaining integrity is showed elsewhere. However, the statement made (before any data is shown) is that rim15 is already known to be important for biotrophic growth. The authors therefore miss the point that the role of Rim15 in biotrophic growth in *M. oryzae* has not been shown prior to this paper.

4) Line 71- As mentioned in the discussion, a-KG is known to be a signal for amino acid sufficiency and activate TORC1. Hence, it's not exactly an unknown signal.

5) Line 97: Figure reference is missing.

6) Throughout the paper, at multiple places a plant response has been mentioned. But what this plant response is, and how a general reader might understand this response looking at the figure/image has not been stated.

7) Line 127-130 and the next para start: the authors first state that they have concluded that Rim15 functions in the TOR pathway, but then later state that their own hypothesis is wrong. This type of paradoxical presentation is seen throughout the paper, where authors first assume something to be the case, but then later on provide data to either prove or disprove it. The paper has to be rewritten to present data in a coherent manner.

8) Even though autophagy might be functional post 40hpi in rim15 cells, biotrophic growth is lost, so at what time point during biotrophic growth is Rim15 dependent activation of autophagy required? Do the authors want to separate these points, or is it irrelevant? Unclear.

9) Figure 6: The GO classification could easily be done separately for upregulated and downregulated gene clusters, to make this more informative.

10) Line 215-216: no data has been shown for glutamate, but still mentioned.

11) Figure 6H: Is there any functionality attached to the change in phosphorylation of these enzymes? A change in phosphorylation without a change in activity might have no consequence on function or output. Without this clarity, these data confuse and don't clarify. Also, glutamine amidotransferase may not be relevant in this context.

12) At 44hpi, autophagy increases. However, a-KG treatment in both rim15 and RIM15 cells leads to decreased autophagy, even though biotrophic growth is not affected. What might be the explanation for this?

13) What might be the reason for ammonia inhibiting RIM15 growth?

14) Line 288: extremely (incorrect spelling)

15) The authors should clarify the directionality of a-KG metabolism, and if glutamine synthesis or utilization is important? It is not clear whether it is a-KG, Glu or Gln or another metabolite that dictates autophagy and the rim15 phenotype rescue.

16) Lines 315-316: incorrect statement. If this was the case, then GS/GOGAT inhibition will not lead to any phenotype. Both GS and GOGAT inhibited cells should ideally have more a-KG levels. However, here a-KG supplementation rescues the phenotype. How is this explained?

17) What might be the role of Rim15 secretion in planta for infection? Even though Rim15 is not nuclear in *M. oryzae*, can the authors comment on whether the fungal or in-planta localization of Rim15 might play a role in regulating biotrophic growth.

18) If a-KG inhibits autophagy, should it not prevent cell to cell movement? This is especially because autophagy is the major determinant of cell to cell movement during biotrophic growth. However, this is a tangent from the purpose of this study.

Reviewer #3 (Remarks to the Author):

Autophagy-dependent TOR reactivation drives fungal growth in living host rice cells
Li, Gong et al.

The authors investigated rice pathogenic fungus, *Magnaporthe oryzae*, and they found that cycles of autophagy modulate TOR reactivation via α -ketoglutarate to support biotrophic growth. Their study is unique and interesting. However, their data about autophagy are not convincing to

support their conclusion. This reviewer recommends the authors to resolve the following problems and to resubmit their manuscript.

Overall comments about autophagy study

The authors mainly used 4 items to study autophagy: deletion of RIM15 gene, deletion of IMP1 gene, amiodarone hydrochloride (AM), and 3-methyladenine (3-MA).

However, RIM15 and IMP1 only indirectly participate in TOR and autophagy, respectively, thus the reviewer strongly recommends the authors to perform ALL autophagy study using ATG mutants, such as Figure 3, 4, 6, and 7. As they mention in L458-9 (page 14), *M. oryzae* genome database is available, they can generate deletion of ATG gene of *M. oryzae*.

In addition, AM and 3-MA are used for mammalian autophagy. Especially, "the antiarrhythmic drug amiodarone (AM) has fungicidal activity against a broad range of fungi." (J Biol Chem 282: 37844-53 (2007)). Does AM have any other effect on *M. oryzae* than induction of autophagy? Therefore, AM is not a suitable drug in this study.

Minor comments

Fig1

It is obvious that *rim15* deletion affects biotrophic growth. However, *rim15* deletion also affects growth on plates (Fig3a, 6g and S1a). Therefore, it is unlikely that TOR-RIM15/autophagy axis is exclusively involved in biotrophic growth.

Fig5c

Autophagic cycling is measured by GFP localization in the vacuole. As the authors show in Fig7e, when GFP-Atg8 is delivered into the vacuole by autophagy, only Atg8 portion is degraded and GFP portion is intact and is accumulated in the vacuole. Thus, GFP localization in the vacuole does not precisely represent temporal change of autophagic activity. For example, the authors checked 4-hour timepoint in Fig5c, and how much of GFP signal/band is decreased in 4 hour if they terminate autophagic activity?

L836 (page 32) " b." should be changed to (c).

Reviewer #4 (Remarks to the Author):

The purpose of this study is to determine how fungal metabolic homeostasis is achieved during biotrophy established during plant infection by the rice blast fungus *Magnaporthe oryzae*. To do this, authors look at the master regulator of growth called target-of-rapamycin (TOR) kinase, that regulates growth in the presence of nutrients and triggers autophagy in starvation conditions. Authors determined that biotrophic growth requires TOR activation-inactivation cycles, triggered by the concentration of the metabolite α -ketoglutarate. The authors identified a serine/threonine kinase called Rim15 which, in other organisms, is required for autophagy induction in response to starvation. In *M. oryzae* Rim15 is required for sustaining biotrophy during infection because of the production of the α -ketoglutarate, as null mutants of the kinase are arrested during biotrophic phase, and this phenotype remediated by α -ketoglutarate addition. By a combination of genetics, cell biology, RNA-seq, metabolomics and phosphoproteomics the study dissects this model, identifies the involvement of the GS-GOGAT cycle in α -ketoglutarate production and unravels a mechanism critical for sustaining biotrophy and the progression of the disease. Therefore, the study really covers an interesting aspect of infection. All together the study shows a comprehensive signalling dissection of the TOR-dependent infection-related program in one of the most important pathogens of rice.

I find this study interesting and with good experimental data. It reveals some of the mechanisms and players associated with the pathogenicity of one of the most important pathogens affecting staple crops. I think that the approaches taken, and most conclusions presented are correct. I have however some reservations in relation to the RNA-seq data, phosphoproteomics and metabolomics that I hope can be explained and clarified. I am providing some comments and

suggestions to be addressed that I hope will strengthen the study.

Major:

- The authors describe α -ketoglutarate metabolite as a previously unknown amino acid sufficiency signal that results from the GS-GOGAT cycle. The authors carry a good set of experiments that include comparative transcriptional analysis, phosphoproteomics and metabolomics to understand why *Δ rim15* responds to α -ketoglutarate. I have however thought that the way that these data/results are presented do not reflect all the potential that these three approaches have in explaining what is happening. At the moment, the way that this is presented (specially figure 6 f and h) does not easily help to understand what is happening biologically. It is hard to pinpoint what occurs with genes associated to GS-GOGAT cycle, the concentration of α -ketoglutarate and the phosphorylation status of GS-GOGAT related genes. I think this section of the manuscript needs to be more elaborated and clarified, with clear panels and graphs (perhaps bar charts, heatmaps, tables...). I understand the authors have done a great job focusing and narrowing down such big datasets into GS-GOGAT cycle, the manuscript lacks the context of other genes and pathways probably found in those datasets.

- As I am finding panel 6f and 6h difficult to interpret. One main and very simple question I have is, α -ketoglutarate metabolite is increased (0.48x) in *Δ rim15* compared to Guy11? Moreover, the relevance of the α -ketoglutarate metabolite comes to importance during infection. In my view, it needs to be shown that α -ketoglutarate metabolite production occurs during infection. Carrying a metabolomics experiment of infected tissue or alternatively perhaps an assay to detect α -ketoglutarate directly during infection would really need to be done. In this way, the question would be, does the production correlate with the time at which biotrophic growth occurs? Also, do plants produce α -ketoglutarate that can be used by *M. oryzae*? How is that specific signal coordinated with other metabolic signals in the plant?

- By the way that the RNA-seq data has been presented in the manuscript, it is hard to understand the thinking process of authors to highlight the importance of the GS-GOGAT cycle. Yet more difficult to relate this to RIM15. There is some information in the legend of Fig6. But I think the reader will benefit if the information in relation to the GS-GOGAT was dissected within the text. Moreover, a deeper explanation of the RNA-seq experiment is needed including what sort of genes were found and how was their expression. Are the 5 genes highlighted in the legend the only ones involved in GS-GOGAT? A heatmap with the differences in expression will help. Could authors dissect more precisely genes related to the GS-GOGAT (and other relevant pathways) and really show how the expression of those is altered?

- Moreover, what other genes show differential gene expression that can be contributing to the phenotype?

- Perhaps I have missed this but, why was 36hpi chosen?

- Also, a phosphoproteomics approach has been carried, but what has been found there? Are those GS-GOGAT genes differentially phosphorylated, are those residues conserved? Was TOR found, what residues, are they conserved with Sc? I can see that the activity of TOR has been inferred indirectly from the phosphorylation of Sch9. It is known and stated by authors that there are some differences from the yeast model. Since The TOR activity measurement is indirect identifying those TOR residues is critical. Moreover, a negative control (e.g. +Rapamycin) should have been included in the WB to really show antibody specificity and relative intensity.

- Also, since authors have carried phosphoproteomics, and considering other sets of published phosphoproteomic results, such as Li et al., 2015, Franck et al. 2015, Have authors explored the putative phosphorylated residues of TOR and associated those to its activity to measure directly and correlate phosphorylation with TOR activity?

- What happens to null mutants of GS-GOGAT-related genes?

- Please confirm the null mutants for Rim15 with a southern blot.

- What sort of p-value had the NLS prediction? Have the authors contemplated the possibility that a sub-population of Rim15 translocates to the nucleus, but it is fairly undetectable? Perhaps deleting the NLS or introducing an NES could help really confirm that Rim15 is not nuclear and very different from ScRim15.

- Quantification of 7a and 7b? how many cells progressed to infect secondary plant cells?

Minor:

- State in Suppl. fig 1 (NT), is non treated
- Micrographs 7c are too dark.

Response to Reviewer's Comments

We are deeply grateful to the editor and reviewers for the thoughtful and insightful comments and suggestions. With your guidance, we have made some considerable changes to the manuscript in order to focus the work on the role of Rim15 in coordinating autophagy and glutaminolysis to fuel biotrophic growth. We have included new data as suggested by the reviewers, and have re-designed many figures to better present our data in a coherent narrative. We have also removed data in order to maintain our focus. We believe this is now a much stronger and more compelling manuscript providing valuable new insights into the molecular mechanisms underlying fungal growth in host plant cells. While our conclusions about the role of Rim15 in biotrophic growth and autophagic cycling remain as before, albeit augmented with new data, our significant new findings (discussed in detail below) are as follows:

1. By knocking out genes required for glutaminolysis, we show both how glutaminolysis is essential for host infection, and how Rim15 controls glutaminolysis via phosphorylation of Gdh2, the last enzyme in the conversion of glutamine to α -ketoglutarate. Expressing a phospho-mimicking version of Gdh2 in $\Delta rim15$ remedies growth on glutamine media and almost fully remedies infection.
2. Following on from the observation in (1) above, using a *M. oryzae* ATG mutant we now show how the provision of α -ketoglutarate by autophagy, via glutaminolysis, is the major role of autophagy during biotrophy. Autophagy during biotrophy is dispensible in the presence of an exogenous supply of α -ketoglutarate.
3. We finally ask why glutamine via α -ketoglutarate is the preferred carbon source during biotrophy and conclude from our new data that, like in some other rapidly proliferating cells, using glutamine for growth preserves glucose for generating reducing equivalents through the pentose phosphate pathway. These activate the antioxidants needed to suppress the host ROS burst and prevent triggering host defenses. We found that treating $\Delta rim15$ -infected cells with glucose suppresses host defense responses but does not stimulate $\Delta rim15$ growth, whereas treatment with α -ketoglutarate remedies both.

Together, we believe this version of the manuscript provides unparalleled insights into the metabolic strategies underpinning pathogen colonization of host cells.

Dear Richard,

Thank you again for submitting your manuscript "Autophagy-dependent TOR reactivation drives fungal growth in living host rice cells" to Nature Communications, and please accept my apologies for the long review process. We have now received reports from 4 reviewers and, after careful consideration, we have

decided to invite a major revision of the manuscript.

As you will see from the reports copied below, the reviewers find your work potentially very interesting but raise important concerns. We find that these concerns limit the strength of the study, and therefore we ask you to address them with additional work and substantial revisions.

Please note that the reviewers have different backgrounds and expertise, and have approached your paper from different angles. Nevertheless, they agree that some of the conclusions are not sufficiently supported by the data presented, and highlight an overall lack of clarity in text and presentation.

We thank the editor and reviewers for the thoughtful reading of our work. This manuscript has been significantly improved by incorporating your suggestions, and we have consequently focused and strengthened our argument that the Ser/Thr protein kinase Rim15 coordinates autophagy and glutaminolysis in *M. oryzae* to drive biotrophy while (as we now show) preserving glucose for plant defense suppression, thus improving our understanding of pathogen metabolic strategies underlying plant disease.

Given that the paper already includes a large amount of data, you may consider following reviewer #2's recommendation of rewriting and restructuring the paper, focusing on some of the conclusions that seem more strongly supported (perhaps Rim15 and its role in biotrophic growth?) and de-emphasizing (or removing?) some aspects or data that might not be essential in a restructured manuscript.

We have followed these suggestions by focusing on the role of Rim15 in biotrophic growth. This includes characterizing glutaminolysis mutants (which shows glutamine metabolism to α -ketoglutarate is essential for biotrophy, but the reverse reaction is not), while removing previous data on Asn1 and Imp1 which we now realize did not add much to the Rim15 conclusions. These data will be published elsewhere. Also, we have removed the RNAseq data. Based on our new genetic results, which include remediating $\Delta rim15$ on plates and plants by expressing a phospho-mimetic version of a downstream glutaminolytic enzyme, we now consider that Rim15 does not likely regulate glutaminolytic gene expression as suggested by the RNAseq data but instead modulates glutaminolytic flux at the enzyme activity level. Changes to glutaminolytic gene expression observed previously is likely an indirect response to the changes in metabolites and is thus misleading.

REVIEWER COMMENTS

Reviewer #1 (Remarks to the Author):

The manuscript submitted by Li et al. report on the association between TOR signalling pathways with fungal invasive growth in M. oryzae. The study produced a series of nice imaging data to demonstrated RIM15, a gene which affects integrity of EIHM, and its association with α -ketoglutarate.

Combining with further analysis, the authors demonstrate Rim15-dependent cycles of autophagic flux liberate α -ketoglutarate as an amino acid-sufficiency signal to trigger TOR reactivation and promote biotrophic growth in host rice cells. Overall, I find this study is of great novelty,

We thank the reviewer for their kind assessment of our data.

however, there are still a number of potential problems with current reported evidence and conclusions that need to be addressed.

Major concerns:

1. According to the results sections written in the manuscripts (line 77-141), the most significant phenotype of compromised Rim15 in M. oryzae, is its abolished colonization after biotrophic growth. The evidences of imaging of Bas4-GFP, and PWL2-RFP is a good indication to elucidate EHIM and exam the integrity of EHIM. However, it is ideal and rigorous to use plant transgenic marker line, given that the EHIM is plant-membrane derived structure. The observation of Bas4-GFP, and PWL2-RFP in leaked host cell, is an indirect way to determine integrity of EHIM, only means the secretary pathways are affected, but not directly indicate a defect in biotrophic growth. Considering the plant transgenic marker line is not easy to obtain, there could be also possible to stain the IH with Trypan blue, which is usually used to quantify live cells by labeling dead cells exclusively. Because live cells have an intact cell membrane, trypan blue cannot penetrate the cell membrane of live cells and enter the cytoplasm. In a dead cell, trypan blue passes through the porous cell membrane and enters the cytoplasm.

We thank you for your thoughtful suggestions. We were unable to obtain plant transgenic marker lines, but we were able to stain infected leaves with Trypan Blue, as suggested. This stain assays for intact plant cell membranes. As now shown in new Figure 2a and described in lines 113-120, at 39 hpi and 44 hpi, over 90 % of WT infected cells are not stained by Trypan blue, indicating living cells with intact membranes. In contrast, 61 % and 90 % of cells infected with $\Delta rim15$ at 39 hpi and 44 hpi, respectively, were stained, suggesting disruption of the EIHM. These numbers match the proportion of $\Delta rim15$ -infected cells with Bas4 in the cytoplasm in Fig. 1d and 1e. When considering that within the rice blast community it is accepted that Bas4 in the cytoplasm indicates a loss of interfacial membrane integrity (see for example Khang et al. 2010), together we conclude that the presence of Bas4 in rice cytoplasm is due to disrupted EIHM and not due to increased secretion.

2. Another point I would like to raise, is that the leakage first invaded cell is not necessarily means its plant defence response or impaired/abnormal invasive growth from pathogen side, especially for the case of *Rim15*. Therefore, the statement of “*Rim15* acted in the TOR-*Imp1*-autophagy signalling cascade downstream of TOR but upstream of *Imp1* at a branch with a separate *Rim15*-controlled pathway mediating ROS scavenging and host innate immunity suppress” is clearly not convincing. Although, the ROS assay experiment was performed, but it is not sufficiently strong to support the assumption about “host innate immunity suppress”. A good practice is using ethanol-killed leaf sheath to perform the invasive growth assay, and see if the mutant could continue its further growth as wild type, or trapped in the first cell. Therefore, I really wonder if authors could re-consider the whole assumptions made in the manuscript, as the imaging observation in this section is the foundation of the whole study.

We thank the reviewer for this analysis of the data and their suggestions which, along with addressing other comments below, have bolstered support for the notion that *Rim15* is required for host immune suppression independently of the TOR-*Imp1* signaling pathway. At this point in the manuscript (summarized in Fig. 2f), we strengthen our conclusion by performing the experiments proposed here and below by this reviewer. As discussed in lines 121-129, new Fig 2b shows how $\Delta rim15$ -infected ethanol-killed cells do not elicit defense responses, but nonetheless, biotrophic growth of the mutant is not remediated. Furthermore, in new Fig. 2c, we now show how diphenyleneiodonium chloride (DPI) treatment (which was not performed previously in this study) suppresses plant defense responses in $\Delta rim15$ -infected cells by inhibiting NADPH oxidases to prevent the host oxidative burst, which otherwise, as previously shown, triggers host defenses (eg. see Li et al. 2020). This data, based on previous work with other antioxidation deficient *M. oryzae* mutants by our group and others (including our publications in Nature Microbiology, as explained in the text,) is sufficient to conclude that $\Delta rim15$ is unable to suppress the host oxidative burst. Thus, we thank the reviewer for stimulating our thought in this direction. However, new Fig. 2c also shows that even if plant defenses are suppressed by DPI treatment, $\Delta rim15$ is unable to grow and biotrophic interfacial membrane integrity is lost. Thus, *Rim15* controls two separate pathways, the autophagy pathway in parallel with TOR-*imp1* signaling, and a separate host immunity suppression pathway (Fig 2f). Later in the manuscript, we elaborate on this role further by showing how the *Rim15*-controlled cycles of autophagy and glutaminolysis allow glutamine via α -ketoglutarate to be used as a fuel for growth while glucose is conserved to provide reducing equivalents via the pentose phosphate pathway for antioxidation and the suppression of host defenses (Fig. 8 and final model in Fig. 9, lines 346-368). Consequently, treatment with glucose inhibits host defense responses in $\Delta rim15$ -infected rice cells but does not remediate growth (Fig. 8), unlike α -ketoglutarate treatment, which remediates both. Together, we conclude that *Rim15* is required for suppressing host defenses via antioxidation by preserving glucose. Using glutamine as a carbon source while preserving glucose for

antioxidation is a metabolic tactic utilized by other rapidly proliferating cells such as T cells and cancer.

3. *The authors concluded that autophagic cycling in IH is Rim15-dependent, this is questionable. This conclusion just deduced from the result that the autophagic cycling is abolished in $\Delta rim15$. The defect in autophagic cycling of $\Delta rim15$ could also possibly be a consequence of host defense-associated inhibition, not absolutely means Rim15 is involved in autophagic cycling. Therefore, I think it is not a solid conclusion.*

The reviewer raises a great point, that the defect in autophagy might not be due directly to the loss of Rim15 control but is rather as consequence of the elevated plant defenses. However, our new results argue against this mainly because, as outlined above, suppressing host defenses by either killing the host cell or treating with DPI was not sufficient to restore biotrophic growth. Therefore, growth is dependent on Rim15 and, as we show later, is also contingent on autophagic cycling, which is abolished in Rim15. When all our data are considered together (as presented in Fig 9), it is safe to assume that Rim15 controls autophagic cycling to promote biotrophy. We might also point out that *RIM15* is involved in autophagy in yeast and, moreover, is required for autophagy in *M. oryzae* in the absence of the plant host (ie on nutrient-free water agar plates, Fig 2e). Moreover, rapamycin treatment (Fig 2d), which activates autophagy independently of Rim15 in yeast, restores membrane integrity (although it is toxic to both WT and $\Delta rim15$ growth).

4. *Line 115-116, "However, about 4 % of $\Delta rim15$ -infected rice cells displayed some Bas4-GFP leakage into the host cytoplasm." (Fig. 1d). How did the authors know that this is a leakage? Could that be a plant side defense response? Because autofluoresce could be found in some incompatibility situation during *M. oryzae* infection (as we have observed). It's lack of evidence to confirm that this situation is a Bas4-GFP leakage into the host cytoplasm.*

Based on our Trypan Blue data, we strongly believe this is Bas4 leakage into the cell. We believe that the erosion of the host membrane is stochastic, such that a small percentage of $\Delta rim15$ -infected cells begin to leak early, and the majority lose membranes at the later time points. Also, it is recognized in the community that Bas4 leakage in the cytoplasm is due to breakage of the membrane (See for example Mosquera et al. 2009, <https://doi.org/10.1105/tpc.107.055228> and Khang et al. 2010, <https://doi.org/10.1105/tpc.109.069666>). Furthermore, autofluorescence would be detected in all the samples, but we do not see this.

5. *I think the multi-omics bioinformatics analyses are really too simple, some important information should be provided. There is an important step of RNA-seq as batch effects removal should be executed before DEGs analysis. Sequence depth of leaf sheath of RNA-seq should be provide, as the ratio of*

plant/pathogen would be significantly high. It is essential to assess the quality of the data carefully, given that the result of PCA is not completely repeated between the replicates. How many genes of M. oryzae are covered from all the samples? Also, the multi-omics data has not been submitted to database yet.

We have removed the RNAseq data for the following reasons. First, we wish to focus the paper on glutaminolysis and autophagy, as recommended by other reviewers and the editor, and thus for space we cannot provide a more in-depth analysis of the RNAseq data here. We plan to publish that data, with the reviewer's recommendations, at a later stage. Second, we initially determined the role of Rim15 in glutaminolysis using plate tests. We originally felt that the RNAseq data lent support to the role of Rim15 in glutaminolysis as it showed altered expression of glutaminolytic genes. However, we now think that this was misleading. In this new version of the paper, we show by functional analysis how glutaminolytic genes are required for infection (lines 187- 292). However, we now also show how Rim15 controls this process not by regulating gene expression, but by phosphorylation of NAD-dependent glutamate dehydrogenase (Gdh2), the last enzyme in the conversion of glutamine to α -ketoglutarate (Fig 4). Therefore, although the RNAseq data showed altered GS-GOGAT gene expression, this could be an indirect effect of losing Rim15, for example a response to the altered metabolites in $\Delta rim15$ strains. Considering our strong Gdh2 phosphorylation data (whereby a phospho-mimicking version of Gdh2 in $\Delta rim15$ remedies the loss of glutaminolysis and pathogenicity in $\Delta rim15$, Fig 4), we decided that the RNAseq data adds nothing to our new focused story, and has thus been removed.

6. It is required to detect the level of α -Ketoglutarate (α -KG) in planta. For example, it is possible to perform an α -Ketoglutarate (α -KG) fluorometric detection assay, which provides a fluorescence-based method for quantifying α -KG.

We detected α -ketoglutarate in rice leaves in an earlier metabolomic study (Wilson et al. 2019), and this is now discussed in the text (line 285 - 292). However, α -ketoglutarate is not sufficiently acquired by the mutants otherwise they would grow without supplementation. Using our new glutaminolysis mutants, we now show how adding high concentrations of exogenous glutaminolytic metabolites remedies $\Delta rim15$ growth via the conversion in the fungal cell to α -ketoglutarate, confirming both that plant glutaminolytic metabolites are not acquired from the host in high quantities, perhaps to avoid detection, and that glutamine is the preferred carbon source for biotrophic growth via its conversion to α -ketoglutarate (Fig 6).

Minors:

1. It would be helpful and great to take the message if the images in the figures could be arranged in a decent way. For instance, Figure 4, Figure 5, Figure 7, Fig S2, Fig S3.

We have made many changes to the figures that we hope now better emphasize the take home message.

2. *Fig. 1b, the lesions are not typical.*

This is now Fig 1a. The lesions for WT and the complement strain have been swapped out with others taken at the same time but which show more typical lesions.

3. *Fig. 3a, needs statistical data.*

Fig 3a is now Fig 2e. Statistical data is included in the source data.

4. *Fig. 3b, scale bar should be in similar place for consistence.*

This has been changed.

5. *Fig. 3b, needs statistical data.*

Fig. 3b is now Fig 2d, and the data has been added as requested.

6. *Fig. 6g, needs statistical data.*

Fig. 6g is now Fig. 4a and the radial growth data is available in the Source Data.

7. *Fig. 10a, needs statistical data.*

Fig. 10a is now Fig. 7f and the statistical data has been added.

Reviewer #2 (Remarks to the Author):

This study has primarily found a critical role for Rim15 in regulating biotrophic growth in M. oryzae. I will first summarize my main take on this paper: the central findings are novel and extremely exciting, and should be published.

We thank the reviewer for their kind words about the work.

I also think this manuscript suffers from too much data (which not just takes away from central points, but creates multiple new tangents). The manuscript suffers terribly because (i) for a large part, the statements, inferences (and title) are distinct from the main findings, (ii) at many parts of the manuscripts,

an inference (or conclusive statement) precedes the actual data shown, and (iii) a lot of data is included that distracts (or actually diverts away from) the main findings.

We thank the reviewer for all their insightful comments. We have revised the paper to focus on the role of Rim15 in autophagy, glutaminolysis and plant defense suppression, and we have provided more data to support our main conclusions, including generating glutaminolysis mutants and showing how Rim15 regulates glutaminolysis by phosphorylation control of Gdh2. Other data, such as the $\Delta asn1$ and $\Delta imp1$ mutants, have been removed for clarity and will be published elsewhere. We have also addressed the issues of conclusive statements preceding the data.

My review below will have two parts, broader comments, and a few very specific comments.

General comments:

1) the title suggests a role for autophagy-dependent TOR reactivation in fungal growth in living host rice cells. While the data presented is excellent, it has not been presented in the form of a coherent story that supports the title. The paper requires rewriting and changing of title and figure panels to support one coherent story. For example, a title of 'Rim15/greatwall kinase drives fungal growth in living host rice cells, controlling autophagic flux' (or something vaguely like that) would be far more appropriate. The data presented currently, and the title are quite different.

We thank the reviewer for these great comments. We agree and, in addition to focusing the paper on the role of Rim15 in coordinating autophagy and glutaminolysis, as discussed in detail below, our new title is:

Fungal Ser/Thr kinase-mediated α -ketoglutarate production via autophagy reactivates TOR and fuels growth in host cells

To focus the paper on one coherent story, we provide the following new data:

1) Lines 187-250: We used the proteomics data following growth on glutamine to identify genes likely involved in glutaminolysis. We deleted *GLT1* involved in converting glutamine to glutamate and *GDH2* involved in converting glutamate to α -ketoglutarate, and show they are both required for glutaminolysis. Specifically, $\Delta glt1$ can use glutamate but not glutamine as a C source, $\Delta gdh2$ cannot use either, and both are impaired for infection. For clarity, we no longer discuss the related GS-GOGAT pathway. We also knocked out *GDH1* involved in the reverse reaction of α -ketoglutarate conversion to glutamate via NH₄ assimilation. It is not affected for glutaminolysis and is fully virulent.

II) Lines 251-292: Using the glutaminolytic mutants, we confirm that glutamine is the preferred carbon source during infection, and that its conversion to α -ketoglutarate in the fungal cell is essential for infection, by showing how treatment with glutamate and α -ketoglutarate but not glutamine remediates $\Delta glt1$ invasive growth while only treatment with α -ketoglutarate remediates $\Delta gdh2$ infection. Thus, glutamine is the preferred carbon source during infection via its metabolism to α -ketoglutarate in the fungal cell.

III) Lines 235-250: Using the phosphoproteomics data, we targeted the ser24 phosphosite on Gdh2 (reduced in abundance in $\Delta rim15$ mycelia during glutaminolysis). First, we show how complementation of $\Delta gdh2$ with a gene expressing a phospho-null version of Gdh2 only partially remediates growth on glutamine media, and does not restore pathogenicity, compared to complementation with a copy of WT *GDH2*. Thus, Ser24 phosphorylation is required for optimal Gdh2 function. Next, we made a mutated form of the *GDH2* gene expressing a phospho-mimetic version of Gdh2 and expressed it in $\Delta rim15$. Expressing phospho-mimetic Gdh2 remediated $\Delta rim15$ growth on CM and glutamine (as C source) media and almost fully restored virulence. We conclude that Rim15 (directly or indirectly) mediates Gdh2 phosphorylation to control glutaminolysis, a process required for rice infection. This is a major role for Rim15 during infection, and might explain why Rim15 is located in the cytoplasm where Gdh2 is also predicted to localize.

IV) The above results convinced us that Rim15 controls glutaminolysis via Ser24 phosphorylation and not at the transcriptional level. Although we originally showed altered expression of glutaminolytic genes in $\Delta rim15$ *in planta*, we now consider those changes to be red herrings, likely the result of altered metabolite repression due to reduced Gdh2 function rather than direct control by Rim15 signaling. Because we determined the role of Rim15 in glutaminolysis from plate tests, and merely used the RNAseq data to bolster support for this notion, we now consider that the RNAseq data adds nothing to our story. In addition, discussion of the other gene expression changes would take up too much space. Thus, we have removed this data entirely from the manuscript. (We have also removed the $\Delta asn1$ and $\Delta imp1$ data, which similarly adds little here).

V) Lines 346-368: The final piece of the puzzle, which explains why adding α -ketoglutarate both suppresses plant defenses in $\Delta rim15$ and suppresses autophagy in $\Delta rim15$ and WT during biotrophic both (two processes we frankly could not reconcile from the data of the previous version of the manuscript without invoking some additional but unknown TOR-dependent pathway involved in plant defense suppression), fell into place when we hypothesized that, like other rapidly proliferating cells, glutamine might be a preferred carbon source to preserve glucose as a source of reducing equivalents via metabolism through the pentose phosphate pathway, rather than being used as a fuel in the TCA cycle. To test our hypothesis, we added glucose to $\Delta rim15$ infected rice cells. This suppressed host plant

defense responses (similar to adding DPI to inhibit the host ROS burst (new Fig 2c)), but it did not restore *Δrim15* biotrophic growth. This was in sharp contrast to α -ketoglutarate treatment, which remediated both. Our conclusion is that glucose is used predominantly for antioxidation but is depleted in *Δrim15* when it is drawn into the TCA cycle for growth instead of α -ketoglutarate, resulting in plant defense elicitation. However, adding α -ketoglutarate preserves glucose for antioxidation and host ROS detoxification (hence no plant defense responses in *Δrim15* treated cells), but glucose is not used to fuel extensive growth.

VI) Lines 328-345: The suppression of autophagy by α -ketoglutarate treatment is consistent with the notion that the major role of autophagy during biotrophy is to provide α -ketoglutarate (via glutaminolysis) as a TCA substrate in order to preserve glucose for antioxidation. This is confirmed using the *M. oryzae* *Δatg8* mutant that cannot colonize wounded leaf tissue and cause disease unless treated with α -ketoglutarate. Thus, when sufficient α -ketoglutarate is provided exogenously, autophagy is dispensible for rice cell colonization (new Fig. 7g).

VII) We conclude that fungal metabolism is controlled in *M. oryzae* by Rim15 during biotrophy to coordinate autophagy with glutaminolysis in order to supply α -ketoglutarate for TOR reactivation and as a fuel for growth while preserving glucose for host plant defense suppression.

2) The authors show a role of Rim15 in regulating biotrophic growth, specially post 39hpi. They then show that this might be because Rim15 regulates autophagy (in rim15 cells, activation of autophagy at 36hpi by AM or Rapa rescues the rim15 phenotype at 44hpi). rim15 cells have a delay in autophagy activation, which suggests Rim15 function is important for temporal activation of autophagy during biotrophic growth. Further, autophagic flux cycles during this growth, which does not happen in a rim15 mutant. However, there are some large gaps in the final conclusions. First, the argument that autophagy cycling is important for biotrophic growth has not been demonstrated (but should be implied), as inhibition of autophagy at any time point leads to a defect in biotrophic growth.

Lines 167-186: We consider that adding 3-MA at any time blocks autophagy in the next cycle, ie abolishes the subsequent peak. We have now shown this is the case by quantifying autophagic activity (as % GFP in vacuoles) after adding 3-MA at the 32 hpi or 40 hpi autophagy activity troughs and finding that the succeeding peaks at 36 hpi and 44 hpi, respectively, were abolished (new Fig 3h).

Further, even though autophagy might be functional post 40hpi in rim15 cells, biotrophic growth is lost, so at what time during biotrophic growth is Rim15 dependent activation of autophagy required?

Rim15-dependent autophagy is required to maintain interfacial membranes after 36 hpi. This is now added to our model in Fig 2f (previous Fig 3c). We also now show that loss of *RIM15* results in reduced

biotrophic growth at 28 hpi, when autophagy is activated in WT but impaired in $\Delta rim15$, thus *RIM15* is required immediately for optimal growth. However, the presence of intact membranes at this early time suggests that basal autophagy (*RIM15* is required for autophagy induction but not basal autophagy) is sufficient to maintain membrane homeostasis until 36 hpi (Lines 182-186). We have also indicated in the text that although autophagy is activated in $\Delta rim15$ at 40 hpi, this is likely too late to remedy the biotrophy defects. Furthermore, because this is Rim15-independent autophagy acting under the redox stress conditions of the host ROS burst and activated plant defenses, it may be the result of stress-induced autophagy (autophagy is activated by many different pathways) or even apoptosis-activated autophagy, consistent with observations in the yeast $\Delta rim15$ mutant. This is now discussed in the text (lines 167-171).

3) rim15 mutants have a differential transcriptional profile with multiple metabolic nodes affected. This in itself is not surprising. However, the authors then focus on one of these nodes - glutamine metabolism. Here, the first argument that GS-GOGAT directly regulated is not convincing at all (based on just gene expression). The authors should hence clarify that the difference is at protein levels. Glt1 (GOGAT) transcripts hardly change, but protein decreases (which implies very different types of regulation). Next, the authors provide data that supports a role for glutamine metabolism, but not really GS-GOGAT. Is GOGAT itself central to their story? Not really, but the presentation makes the case that it is. Metabolic profiling shows a decrease in steady-state Glu and α -KG levels in rim15 cells.

We agree that the previous emphasis on GS-GOGAT was misleading. We have now focused, with functional analysis, on glutaminolysis. We first determined a role for Rim15 in glutaminolysis based on plate tests, and the RNAseq data appeared to support that notion. However, we now think that this was misleading as we have gone on to show here how Rim15 controls glutaminolysis by phosphorylation control of Gdh2, the last enzyme in the glutaminolytic pathway. Also, because expressing a phospho-mimetic version of Gdh2 in $\Delta rim15$ remediates infection (Fig 4), Rim15 does not likely control gene expression directly. Therefore, not only have we removed the RNAseq data for space, but we have also ignored the GS-GOGAT genes that were identified in the RNAseq data and instead focused only on those genes involved in glutaminolysis. Candidate glutaminolytic genes were identified from the proteomic and phosphoproteomic data sets, which were generated under glutaminolytic conditions. We then knocked out the genes and go on to show how genes involved in metabolizing glutamine to α -ketoglutarate, but not the reverse reaction, are essential for rice blast infection (Fig 4). Moreover, remediation of infection by supplementation is dependent on fungal gene function such that $\Delta rim15$, which is impaired but not abolished for glutaminolysis, can be remediated by excess glutamine, glutamate and α -ketoglutarate; Glt1, which is required for the conversion of glutamine to glutamate, can be remediated by glutamate and α -ketoglutarate but not by glutamine; and Gdh2, which converts glutamate to α -ketoglutarate, can only be remediated by α -ketoglutarate (Fig 6).

Further, rim15 mutants are not able to utilize glutamine effectively irrespective of presence or absence of glucose. The authors should clarify that the growth defect in rim15 is also seen in presence of glucose.

Lines 188- 197: We have clarified this with regards to the plate tests. In our new images it is clear that $\Delta rim15$ grows less and morphologically distinct to WT on plates with glutamine and glutamate as C source, but morphologically similar (though still reduced) compared to WT on all other media. Further, Fig 8 now shows how treatment of $\Delta rim15$ -infected rice cells with glucose suppressed host defenses (by supplying reducing equivalents for antioxidation), but did not remediate $\Delta rim15$ biotrophic growth. Thus, although $\Delta rim15$ grows better (ie morphologically similar to WT, though with some reduced radial growth) on plates with glucose and glutamine or glutamate as nitrogen sources compared to glutamine or glutamate alone, *in planta*, the harsh plant cell environment necessitates the utilization of glucose for antioxidation and growth is not restored by glucose.

4) Hence, the argument is that rim15 phenotype of decreased biotrophic growth is due to decreased a-KG in rim15 cells, which is consistent with an a-KG growth rescue. However, here the authors suggest that this is due to a rescue of autophagy suppression at 44hpi (by adding a-KG at 36hpi). However, at 44hpi autophagy is highest during biotrophic growth (Fig. 5C). So, the logic of why the suppression of autophagy would rescue the phenotype is entirely unclear. How do the authors explain this? A reinterpretation of the data would be warranted. Along these lines, does a-KG addition at 36hpi affect biotrophic growth at 44hpi in RIM15+ cells since autophagy is affected?

Lines 308-368: This was a wonderful point made by the reviewer that forced us to think deeply about this paradox, which could not have been resolved without the glucose treatment data discussed above. In essence, we now believe that the major or even sole role of Rim15-dependent autophagy is to supply α -ketoglutarate via glutaminolysis to fuel biotrophic growth in order to preserve glucose for antioxidation. Adding exogenous α -ketoglutarate obviates this need and growth occurs without autophagy induction. This is consistent both with our final model (Fig. 9) and with the notion that Rim15 function is directed towards supporting growth while maintaining antioxidation-mediated host defense suppression. Furthermore, α -ketoglutarate addition to $RIM15^+$ at 36 hpi did not affect biotrophic growth at 44 hpi even though autophagy was suppressed. We additionally show here how an ATG mutant abolished for autophagy, $\Delta atg8$, can colonize wounded leaf tissues ($\Delta atg8$ appressoria are non-functional) only if treated with exogenous α -ketoglutarate. Thus, we provide conclusive evidence that autophagy is dispensible for biotrophy and disease development in the presence of exogenous α -ketoglutarate. This has now been specifically discussed in the text.

5) Even though S6Kinase phosphorylation increases post addition of a-KG in water, it is not clear how

this is related to/can be contextualized w.r.t biotrophic growth. What appears to be the case is that a-KG has a metabolic role, rather than signalling role.

The reviewer is correct. We now discuss that α -ketoglutarate has a metabolic role as a preferred carbon source that supplants glucose for fuel. This is in addition to a signaling role for TOR activation, based on the western blot analysis under nutrient-free conditions (lines 305-306 and lines 362-367).

Further, a-KG addition activating TOR should result in less autophagy and directly contradicts the starting model of autophagy activation at 36hpi (and even TOR inhibition by Rapa) being required for biotrophic growth at 44hpi, in context of rim15 mutants. How do the authors clarify these opposing data?

Good points. We can now articulate that our results are consistent with α -ketoglutarate supplementation at 36 hpi abolishing autophagy induction in WT, and suppressing autophagy through at least 44 hpi, as the presence of α -ketoglutarate obviates the need for autophagy (lines 293-345).

And again, the title of the paper is not justified from the data. Going by their data, the role of autophagy dependent TOR activation is not the focus of the paper (and Rim15's role certainly is).

We have changed the title to reflect that Rim15-dependent α -ketoglutarate production is the focus of the paper.

6) Relatedly, glutamine metabolism is important for regulating biotrophic growth. Consistent with this, Gln/Glu addition in rim15 mutants also rescue the phenotype. All these data does not differentiate between metabolic or signalling roles for a-KG being important. The authors have overemphasized a signalling role for a-KG. Further, a-KG levels will ideally be high in GS or GOGAT inhibited cells, as a-KG is either a direct or an indirect substrate for both enzymes, So why will inhibition of GS/GOGAT lead to a rim15 like phenotype?

Lines 218-234: Thank you for these valuable insights. These comments motivated us to functionally characterize fungal genes required for metabolizing glutamine to α -ketoglutarate, as described above. These mutants, and their remediation by Gln and/ or Glu and/or a-KG, indicate a biotrophic role for α -ketoglutarate that is dependent on fungal metabolism. For the purpose of providing more concrete evidence and a more coherent story, the inhibition data (which could include off-target effects) has been removed and replaced with the genetic data. However, our previous conclusions about the inhibition data are inline with our new data: GS or GOGAT inhibited cells consume α -ketoglutarate in the TCA cycle until it is depleted, then consume glucose, leading to growth inhibition.

7) Figure 9 is a complete diversion from the main topic of 'how might Rim15 be playing a role in regulating biotrophic growth'. It is not clear how these data fit into this story at all. If the authors choose to keep it here, they must explain why Asn addition in *asn1* mutant rescues biotrophic growth, but does not do so in *rim15* cells, and why in these cells activation of autophagy by AM is not rescuing the phenotype? This appears to be a major (and unconnected) diversion of this study.

The reviewer is correct, and this data has been removed.

8) The final data suggest that even during autophagy inhibition, addition of α -KG rescues biotrophic growth in WT cells. This argues that α -KG (or an immediate metabolic product of α -KG, glutamate or succinate) is the reason why autophagy is required. The paper ends with a circular argument of α -KG being needed at 36hpi for autophagy inhibition which rescues *rim15* phenotype, but simultaneously autophagy activation at 36hpi is important for α -KG production. Further, as mentioned previously, in WT cells, autophagy is highest at 44hpi, not inhibited. How can the authors explain this incongruity?

We hope the reviewer is satisfied with our explanation that, like with $\Delta rim15$, adding exogenous α -ketoglutarate to WT suppresses autophagy through TOR activation because the major function of autophagy is to supply α -ketoglutarate for metabolism and to signal growth.

Some more specific comments:

1) Figure panels should be changed, so that all data supporting one argument can all be included in one single figure. The data supporting the same points seem to be divided between figures.

This has been done.

2) All throughout papers, panels having separate data should be called out separately, and must state what they say. E.g. Supplementary fig 1D and figure 6.

This has been done.

3) Lines 64-66: the data for these proteins being important for maintaining integrity is shown elsewhere. However, the statement made (before any data is shown) is that *rim15* is already known to be important for biotrophic growth. The authors therefore miss the point that the role of Rim15 in biotrophic growth in *M. oryzae* has not been shown prior to this paper.

This section of the paper has been re-written.

4) Line 71- As mentioned in the discussion, a-KG is known to be a signal for amino acid sufficiency and activate TORC1. Hence, it's not exactly an unknown signal.

This has been re-written. In the discussion, we made a mistake in the previous version and have now clarified that this is a TOR re-activating signal. TOR reactivation is a less studied phenomenon than TOR activation.

5) Line 97: Figure reference is missing.

Thank you, this has been corrected.

6) Throughout the paper, at multiple places a plant response has been mentioned. But what this plant response is, and how a general reader might understand this response looking at the figure/image has not been stated.

The plant response is a defense response visible by eye as granules and as increased ROS by DAB staining. This has been clarified throughout.

7) Line 127-130 and the next para start: the authors first state that they have concluded that Rim15 functions in the TOR pathway, but then later state that their own hypothesis is wrong. This type of paradoxical presentation is seen throughout the paper, where authors first assume something to be the case, but then later on provide data to either prove or disprove it. The paper has to be rewritten to present data in a coherent manner.

We now have included new data to show how suppressing plant defense responses does not remediate biotrophic growth, thus better illustrating how Rim15 works in a plant defense pathway separate from its role in autophagy (this is shown later in the manuscript to be in order to preserve glucose for antioxidation). Therefore, these specific lines and paragraphs have been re-written. The rest of the paper has been re-written as suggested to present the data in a coherent manner.

8) Even though autophagy might be functional post 40hpi in rim15 cells, biotrophic growth is lost, so at what time point during biotrophic growth is Rim15 dependent activation of autophagy required? Do the authors want to separate these points, or is it irrelevant? Unclear.

Rim15-dependent autophagy is required at 28 hpi for optimal growth and post 36 hpi for membrane integrity, as discussed above. Autophagy after 40 hpi is Rim15-independent and might be due to stress-activated autophagy or, as seen in yeast $\Delta rim15$ mutants, apoptosis. These points have now been made more clearly in the manuscript (lines 167-186).

9) *Figure 6: The GO classification could easily be done separately for upregulated and downregulated gene clusters, to make this more informative.*

To focus the paper, we have removed the RNAseq data, which added nothing to this story, as discussed above.

10) *Line 215-216: no data has been shown for glutamate, but still mentioned.*

This is now shown in fig. 4a along with the growth data for the glutaminolytic mutants we have characterized.

11) *Figure 6H: Is there any functionality attached to the change in phosphorylation of these enzymes? A change in phosphorylation without a change in activity might have no consequence on function or output. Without this clarity, these data confuse and don't clarify. Also, glutamine amidotransferase may not be relevant in this context.*

Lines 235-250: We thank the reviewer for this fabulous suggestion to clarify this data. We have removed the GS-GOGAT data and only focused on genes involved in glutaminolysis and the reverse reaction, as described above. In addition to deletion analysis, we assessed the physiological relevance of the Ser 24 phosphosite on Gdh2, the enzyme that catalyzes the conversion of glutamate to α -ketoglutarate. Whereas complementation of $\Delta gdh2$ with the WT copy of GDH2 restored growth on glutamine and glutamate and remediated infection, complementation with the phospho-null version of Gdh2 only partially remediated $\Delta gdh2$ on media and did not remediate infection. Thus, Ser24 phosphorylation is required for Gdh2 function. Most importantly for our investigation of Rim15 function, expressing a phospho-mimetic version of Gdh2 in $\Delta rim15$ remediated growth on glutamine and glutamate media and almost fully restored infection. Thus, we conclude that phosphorylation control of Gdh2 by $\Delta rim15$ is critical for infection (see Fig 4). Indeed, Gdh2 is likely a major downstream target of Rim15.

12) *At 44hpi, autophagy increases. However, a-KG treatment in both rim15 and RIM15 cells leads to decreased autophagy, even though biotrophic growth is not affected. What might be the explanation for this?*

As stated above, we believe this is consistent with the role of autophagy being to provide α -ketoglutarate during biotrophy to fuel growth while preserving glucose for antioxidation. Exogenous α -ketoglutarate suppresses autophagy and thus suggests that autophagy per se is not required for biotrophic growth, rather it is the provision of α -ketoglutarate that is critical.

13) *What might be the reason for ammonia inhibiting RIM15 growth?*

Line 269: Based on the new mutant data that shows glutaminolysis but not the reverse NH_4 assimilating pathway is required for infection, we now think that it is simply because NH_4 assimilation does not contribute to α -ketoglutarate production.

14) *Line 288: extremely (incorrect spelling)*

Thank you. However, this section has now been re-written and the word removed entirely.

15) *The authors should clarify the directionality of a-KG metabolism, and if glutamine synthesis or utilization is important? It is not clear whether it is a-KG, Glu or Gln or another metabolite that dictates autophagy and the rim15 phenotype rescue.*

Great suggestion! This spurred us to knock out genes required for glutaminolysis and one required for the reverse reaction. The glutaminolytic genes are required for infection and are remediated by downstream glutaminolytic metabolites. The reverse reaction is not required for infection. These results are summarized in Fig. 4e.

16) *Lines 315-316: incorrect statement. If this was the case, then GS/GOGAT inhibition will not lead to any phenotype. Both GS and GOGAT inhibited cells should ideally have more a-KG levels. However, here a-KG supplementation rescues the phenotype. How is this explained?*

As mentioned above, this data has been removed as there could be off-target effects of the inhibitors and, moreover, the genetic analysis provides more solid data.

17) *What might be the role of Rim15 secretion in planta for infection? Even though Rim15 is not nuclear in M. oryzae, can the authors comment on whether the fungal or in-planta localization of Rim15 might play a role in regulating biotrophic growth.*

As now specified in the text, we believe the major role of Rim15 is to control Gdh2 phosphorylation and propose that this occurs in the cytoplasm. We do not consider that Rim15 is secreted.

18) If α -KG inhibits autophagy, should it not prevent cell to cell movement? This is especially because autophagy is the major determinant of cell to cell movement during biotrophic growth. However, this is a tangent from the purpose of this study.

Thank you for raising this issue. We have given this paradox great thought and, based on our new experiments showing glucose is required to suppress host defenses but not fuel *M. oryzae* growth, we propose the following: the main role of autophagy is to supply α -ketoglutarate to fuel growth in order to preserve glucose for antioxidation. In the presence of exogenous α -ketoglutarate, autophagy is not required. Thus, autophagy is a major determinant of cell-to-cell growth but only in so far as it is required to supply α -ketoglutarate as a preferred carbon source.

Reviewer #3 (Remarks to the Author):

Autophagy-dependent TOR reactivation drives fungal growth in living host rice cells
Li, Gong et al.

The authors investigated rice pathogenic fungus, Magnaporthe oryzae, and they found that cycles of autophagy modulate TOR reactivation via α -ketoglutarate to support biotrophic growth. Their study is unique and interesting.

We are grateful to the reviewer for their kind appraisal of the work.

However, their data about autophagy are not convincing to support their conclusion. This reviewer recommends the authors to resolve the following problems and to resubmit their manuscript.

Overall comments about autophagy study

The authors mainly used 4 items to study autophagy: deletion of RIM15 gene, deletion of IMP1 gene, amiodarone hydrochloride (AM), and 3-methyladenine (3-MA).

However, RIM15 and IMP1 only indirectly participate in TOR and autophagy, respectively, thus the reviewer strongly recommends the authors to perform ALL autophagy study using ATG mutants, such as Figure 3, 4, 6, and 7. As they mention in L458-9 (page 14), M. oryzae genome database is available, they can generate deletion of ATG gene of M. oryzae.

Lines 377-345: We thank the reviewer for this great suggestion. There are some difficulties with performing all the studies using ATG mutants in *M. oryzae*, mainly because *M. oryzae* Δatg mutants do not form functional appressoria like $\Delta rim15$. However, following the spirit of the reviewer's suggestion, we

have performed a key experiment with a key ATG mutant that, we believe, confirms the importance and function of autophagy during biotrophic growth. *M. oryzae* $\Delta atg8$, a mutant in our collection from the Talbot lab, is unable to make functional appressoria, but we used wounded leaves to show for the first time how this mutant is also unable to establish infection in the host. However, like for $\Delta rim15$, treatment with α -ketoglutarate remediated $\Delta atg8$ infection of wounded leaves (Fig. 7f). These results confirm that autophagy is required for biotrophy and support our conclusion, based on new results described above, that autophagy during biotrophy is mainly dedicated to the provision of α -ketoglutarate for growth and is thus dispensable in the presence of exogenous α -ketoglutarate.

In addition, AM and 3-MA are used for mammalian autophagy. Especially, "the antiarrhythmic drug amiodarone (AM) has fungicidal activity against a broad range of fungi." (J Biol Chem 282: 37844-53 (2007)). Does AM have any other effect on M. oryzae than induction of autophagy? Therefore, AM is not a suitable drug in this study.

We have removed the AM data. Although we are confident that AM stimulates biotrophy, as shown in our previous publication using sublethal concentrations (Sun et al. 2018), we do not know how AM treatment affects plant defenses. Because a new focus of the re-written study was to better determine how Rim15 controls plant defenses (we now know it is via its role in preserving glucose for antioxidation), we did not want this treatment data to be misleading. Removing the AM data does not affect our conclusions.

Minor comments

Fig1

It is obvious that rim15 deletion affects biotrophic growth. However, rim15 deletion also affects growth on plates (Fig3a, 6g and S1a). Therefore, it is unlikely that TOR-RIM15/autophagy axis is exclusively involved in biotrophic growth.

Lines 235-250: We hope that our new data, including complete remediation of the $\Delta rim15$ phenotype on plates and partial remediation on plants by expression of a phospho-mimetic version of Gdh2, will convince the reviewer that Rim15 controls autophagy and glutaminolysis to provide α -ketoglutarate for TOR signaling and biotrophic growth. Our conclusion is that the coordination of autophagy and glutaminolysis by Rim15 is the major role of this kinase during biotrophy and is required to supply α -ketoglutarate to fuel growth and activate TOR while preserving glucose for antioxidation-mediated host plant defense suppression.

Fig5c

Autophagic cycling is measured by GFP localization in the vacuole. As the authors show in Fig7e, when GFP-Atg8 is delivered into the vacuole by autophagy, only Atg8 portion is degraded and GFP portion is

intact and is accumulated in the vacuole. Thus, GFP localization in the vacuole does not precisely represent temporal change of autophagic activity. For example, the authors checked 4-hour timepoint in Fig5c, and how much of GFP signal/band is decreased in 4 hour if they terminate autophagic activity?

Lines 174-186: We are deeply grateful to the reviewer for this great idea. We have now checked and shown that terminating autophagy with 3-MA at 32 hpi or 40 hpi abolished autophagy 4 hours later (new Fig. 3h). Thus, the peaks of autophagy we observe are due to dynamic autophagic activity, and are not simply due to the gradual and stable accumulation of GFP.

L836 (page 32) " b." should be changed to (c).

Thank you. This figure has been changed to Fig 3, but we used the original figure legends and needed to catch that typo.

Reviewer #4 (Remarks to the Author):

*The purpose of this study is to determine how fungal metabolic homeostasis is achieved during biotrophy established during plant infection by the rice blast fungus *Magnaporthe oryzae*. To do this, authors look at the master regulator of growth called target-of-rapamycin (TOR) kinase, that regulates growth in the presence of nutrients and triggers autophagy in starvation conditions. Authors determined that biotrophic growth requires TOR activation-inactivation cycles, triggered by the concentration of the metabolite α -ketoglutarate. The authors identified a serine/threonine kinase called Rim15 which, in other organisms, is required for autophagy induction in response to starvation. In *M. oryzae* Rim15 is required for sustaining biotrophy during infection because of the production of the α -ketoglutarate, as null mutants of the kinase are arrested during biotrophic phase, and this phenotype remediated by α -ketoglutarate addition. By a combination of genetics, cell biology, RNA-seq, metabolomics and phosphoproteomics the study dissects this model, identifies the involvement of the GS-GOGAT cycle in α -ketoglutarate production and unravels a mechanism critical for sustaining biotrophy and the progression of the disease. Therefore, the study really covers an interesting aspect of infection. All together the study shows a comprehensive signalling dissection of the TOR-dependent infection-related program in one of the most important pathogens of rice.*

Thank you for your thoughtful summation of our work.

I find this study interesting and with good experimental data.

Many thanks!

It reveals some of the mechanisms and players associated with the pathogenicity of one of the most important pathogens affecting staple crops. I think that the approaches taken, and most conclusions presented are correct. I have however some reservations in relation to the RNA-seq data, phosphoproteomics and metabolomics that I hope can be explained and clarified. I am providing some comments and suggestions to be addressed that I hope will strengthen the study.

Major:

- The authors describe α -ketoglutarate metabolite as a previously unknown amino acid sufficiency signal that results from the GS-GOGAT cycle. The authors carry a good set of experiments that include comparative transcriptional analysis, phosphoproteomics and metabolomics to understand why $\Delta rim15$ responds to α -ketoglutarate. I have however thought that the way that these data/results are presented do not reflect all the potential that these three approaches have in explaining what is happening. At the moment, the way that this is presented (specially figure 6 f and h) does not easily help to understand what is happening biologically. It is hard to pinpoint what occurs with genes associated to GS-GOGAT cycle, the concentration of α -ketoglutarate and the phosphorylation status of GS-GOGAT related genes. I think this section of the manuscript needs to be more elaborated and clarified, with clear panels and graphs (perhaps bar charts, heatmaps, tables...). I understand the authors have done a great job focusing and narrowing down such big datasets into GS-GOGAT cycle, the manuscript lacks the context of other genes and pathways probably found in those datasets.

Lines 187-292: Thank you for these thoughtful comments. The reviewer is correct that the data in original Fig. 6 did not do a good enough job of explaining the observed outcomes. We first determined that $\Delta rim15$ was impaired in glutaminolysis using plate growth tests (which have been extended in this version of the paper, see new Fig. 4a). That led us to perform the proteomics and phosphoproteomics under glutamine-metabolizing conditions. It also led us to conclude, prematurely as it turns out, that the RNAseq data supported this notion of Rim15-dependent glutaminolysis. However, we now think the conclusions from the RNAseq data were not safe and, along with addressing other comments above regarding the need to focus the work, we have now removed the RNAseq data. One reason is that although we saw transcript changes in the *in planta* RNAseq data consistent with a role for Rim15 in controlling glutaminolysis at the gene expression level, we now show that Rim15 controls glutaminolysis via Gdh2 phosphorylation, and a version of Gdh2 carrying a phospho-mimetic amino acid change fully remediates $\Delta rim15$ on glutamine media. Thus, we now consider that the gene expression changes are indirectly the result of alterations in glutaminolytic metabolite concentrations and not due to direct Rim15 control. Please also note that in this version of the paper, we do not consider the GS-GOGAT pathway, which is operational under steady-state conditions, but instead focus on glutaminolysis and those genes that we experimentally determine are required for glutamine metabolism to α -ketoglutarate via glutamate (Fig 4

and 6). In conclusion, we have extended the section of the manuscript previously covered in Fig. 6 to span three figures (new Fig 4-6), while removing some data we do not think adds to the story. We hope this addresses this reviewers' concerns.

- As I am finding panel 6f and 6h difficult to interpret. One main and very simple question I have is, α -ketoglutarate metabolite is increased (0.48x) in $\Delta rim15$ compared to Guy11?

Actually, it is decreased 0.47 x, ie by almost half, in $\Delta rim15$ compared to WT. This is clarified in the text (line 254).

*Moreover, the relevance of the α -ketoglutarate metabolite comes to importance during infection. In my view, it needs to be shown that α -ketoglutarate metabolite production occurs during infection. Carrying a metabolomics experiment of infected tissue or alternatively perhaps an assay to detect α -ketoglutarate directly during infection would really need to be done. In this way, the question would be, does the production correlate with the time at which biotrophic growth occurs? Also, do plants produce α -ketoglutarate that can be used by *M. oryzae*? How is that specific signal coordinated with other metabolic signals in the plant?*

Lines 275-292: We have now performed new experiments that we hope clarifies the role of α -ketoglutarate. First, in a previous publication (Wilson et al. 2019), we detected α -ketoglutarate in rice leaves, and so it is indeed present in the host. We have mentioned that data here. However, α -ketoglutarate is not available to the fungus in amounts that can suppress the biotrophic defects of $\Delta rim15$, which requires 10 mM of α -ketoglutarate, otherwise we would not see any growth defects of the mutant. Rather, during WT infection, α -ketoglutarate is generated internally by the fungus and is not associated with the plant unless we add it exogenously. We are confident of this because we have now made two additional mutants impaired in glutaminolysis, $\Delta glt1$ which cannot convert glutamine to glutamate, and $\Delta gdh2$, which cannot convert glutamate to α -ketoglutarate. Both mutants are impaired in biotrophic growth. $\Delta glt1$ biotrophic growth can be remediated by exogenous treatment with glutamate and α -ketoglutarate but not glutamine. $\Delta gdh2$ can only be remediated by α -ketoglutarate. We conclude that α -ketoglutarate is a fungal cell-intrinsic metabolite generated during infection to support biotrophic growth.

- By the way that the RNA-seq data has been presented in the manuscript, it is hard to understand the thinking process of authors to highlight the importance of the GS-GOGAT cycle. Yet more difficult to relate this to RIM15. There is some information in the legend of Fig6. But I think the reader will benefit if the information in relation to the GS-GOGAT was dissected within the text. Moreover, a deeper explanation of the RNA-seq experiment is needed including what sort of genes were found and how was their expression. Are the 5 genes highlighted in the legend the only ones involved in GS-GOGAT? A

heatmap with the differences in expression will help. Could authors dissect more precisely genes related to the GS-GOGAT (and other relevant pathways) and really show how the expression of those is altered?

We have omitted the RNAseq data. However, we now provide functional evidence for the glutaminolytic genes discovered in the proteomics dataset (lines 218-232), which is a more robust analysis than could be inferred from the RNAseq data. Furthermore, expressing in $\Delta rim15$ a phospho-mimicking version of Gdh2, an enzyme required for glutaminolysis, remediates $\Delta rim15$ growth on glutamine media and almost fully restores virulence, thus more carefully relating these genes to *RIM15*. We conclude glutaminolysis is important for infection and controlled by Rim15 via phosphorylation of Gdh2 (lines 235-250).

- Moreover, what other genes show differential gene expression that can be contributing to the phenotype?

We have not focused on differential gene expression. However, because a phospho-mimetic version of Gdh2 produced in $\Delta rim15$ remediates the $\Delta rim15$ phenotype, we can conclude that Gdh2, converting glutamate to α -ketoglutarate, contributes significantly to the $\Delta rim15$ phenotype.

- Perhaps I have missed this but, why was 36hpi chosen?

36 hpi was chosen because in a previous study, we found this was optimal for applying treatments because it allowed some establishment of IH. Here, we have used other time points too, such as 32hpi and 40 hpi for 3-MA treatment.

- Also, a phosphoproteomics approach has been carried, but what has been found there? Are those GS-GOGAT genes differentially phosphorylated, are those residues conserved? Was TOR found, what residues, are they conserved with Sc? I can see that the activity of TOR has been inferred indirectly from the phosphorylation of Sch9. It is known and stated by authors that there are some differences from the yeast model. Since The TOR activity measurement is indirect identifying those TOR residues is critical.

We thank the reviewer for these comments. To focus the work on glutaminolysis, we now have a new section explaining what glutaminolytic proteins were found in the proteomics dataset. TOR kinase is in the dataset but is not altered in abundance in $\Delta rim15$ compared to WT, but that is expected because Rim15 is acting in parallel to TOR signaling. Furthermore, our study of the datasets is, due to space considerations, not exhaustive. However, we do show how a Rim15-dependent phosphorylated residue on Gdh2 found in the phosphoproteomics data is critical for function. Moreover, expressing Gdh2 with a phospho-mimetic amino acid site at this residue in $\Delta rim15$ remediates glutaminolysis and infection,

indicating that phosphorylation of Gdh2 is a major role for Rim15 (lines 235-250). Thus, this prompting by the reviewer for a more detailed examination of the data has revealed profound new insights into the control of the infection process.

Moreover, a negative control (e.g. +Rapamycin) should have been included in the WB to really show antibody specificity and relative intensity.

Thank you for this suggestion, we have repeated the Westerns with Rapamycin controls (Fig. 7d,e).

- Also, since authors have carried phosphoproteomics, and considering other sets of published phosphoproteomic results, such as Li et al., 2015, Franck et al. 2015, Have authors explored the putative phosphorylated residues of TOR and associated those to its activity to measure directly and correlate phosphorylation with TOR activity?

We have not done this because in the glutamine media we are using to assess putative Rim15 targets, we expect TOR to be active in both samples.

- What happens to null mutants of GS-GOGAT-related genes?

Lines 218-292: Great question and a major motivation for us to generate glutaminolysis mutants. As noted above, glutaminolysis mutants are impaired in biotrophy and this is remediated by supplementation with glutaminolytic metabolites downstream of the targeted enzyme.

- Please confirm the null mutants for Rim15 with a southern blot.

We thank the reviewer for this suggestion. We were remiss in not including the PCR data that confirms the gene knockouts created in the manuscript, and this has now been added as Supplementary Figure 6. However, we regret that we cannot perform radiolabeled southern blot analyses at the University of Nebraska. Indeed, eighteen of RAW's papers since 2007 have not included southern blots to confirm knockouts. Publications such as *Nature Microbiology* and *PNAS* have accepted that our method (prevalent in the *M. oryzae* community) of knockout confirmation by PCR and phenotypic analysis of several independent mutants followed by complementation, is a rigorous method to confirm the phenotype of a gene deletion. All mutants here were complemented. Therefore, we graciously request that this reviewer please consider precedent when we state that we cannot perform radiolabeled southern blot analysis in RAW's lab, but we are nonetheless extremely confident of our results, and the confirmatory PCRs are now included.

- What sort of p-value had the NLS prediction? Have the authors contemplated the possibility that a sub-population of Rim15 translocates to the nucleus, but it is fairly undetectable? Perhaps deleting the NLS or introducing an NES could help really confirm that Rim15 is not nuclear and very different from ScRim15.

Thank you for this suggestion. We have not performed deleting the NLS or NES because, on more careful inspection, NLSs were not detected in Rim15 when several commonly used NLS prediction programs were employed, including PredictNLS, NLStradamus, and cNLS Mapper. Although PSORT II detected a few short stretches of amino acids enriched in positively charged residues, they are not classical monopartite or bipartite NLS, or even experimental confirmed non-classical NLSs. Therefore, we have removed the information about the Rim15 NLS, and we did not conduct further research on this. However, based on our new data showing Gdh2 is a major target of Rim15, the localization of Rim15 is consistent with that of Gdh2, which is expected to be in the cytoplasm (lines 249-250).

- Quantification of 7a and 7b? how many cells progressed to infect secondary plant cells?

These panels are now Fig. 6b and Fig. 7a, respectively. The quantification data has been added to the images and is available in the Source Data.

Minor:

- State in Suppl. fig 1 (NT), is non treated

Done!

- Micrographs 7c are too dark.

These have been brightened (they are now Figs 7a and b).

REVIEWERS' COMMENTS

Reviewer #1 (Remarks to the Author):

The authors have well addressed all of my concerns.

Reviewer #2 (Remarks to the Author):

The authors have done a commendable job in their revision. The revised manuscript is also now very readable, and makes its key point on the role of Rim15 in biotrophic growth, and balancing autophagic flux. The new data really clarify the role of α -KG (and directions of utilization), and its metabolic importance. I think this study is very exciting as a first report, and also sets up multiple new directions of future inquiry.

Reviewer #3 (Remarks to the Author):

The authors answer most of questions and comments from Reviewer #3. The data they presented defend their conclusion, and this reviewer agrees their manuscript for publication in Nature communications.

Reviewer #4 (Remarks to the Author):

In this study Wilson's laboratory report findings about the role of an uncharacterized serine/threonine protein kinase called Rim15 required for invasive growth of the rice blast fungus. The authors found that Rim15 is required for sustained biotrophy. Moreover Rim15 is necessary for autophagic-flux cycles that liberate α -ketoglutarate (through glutaminolysis) and that reactivate TOR to drive biotrophy. This Rim15-mediated mechanism operates via glutaminolysis (and three enzymes involved and identified in this manuscript).

I find this version of this manuscript better than the previous one. The RNA-seq dataset was not necessary as it is shown now with this new version of the manuscript. Now by having explored the phosphoproteomics and proteomics datasets, it has been precisely shown the involvement of glutaminolysis and the role of three new players Gdh2, Gdh1 and Glt1 in invasive growth. Moreover, a specific serine residue (S24) in Gdh2 has been identified which seems critical as differences between phosphomimetic and phosphodead alleles have been found. This has given a new mechanistic nature for the pathway and the story of this manuscript. I find this new version of the manuscript improved from the previous one with much clearer story and focused findings. I have just some minor comments as authors have addressed most of my comments from the previous revision.

Minor

- Now that the residue S24 has been identified, is there any lead/evidence/experiment to show that Rim15 is the upstream kinase required for the phosphorylation at that residue? Is this residue conserved in other fungi?
- In the model (Fig 9) it will be misleading to show with a continuous arrow the relationship

between Rim15 kinase which will imply (could be interpreted as) direct phosphorylation. Include a discontinued arrow line.

-In the supplementary table 1, no MGG numbers are shown. It is essential that around L201 where Gdh2, Gdh1 and Glt1 are identified, that UniProt codes are given to be able to easily identify them on the Supplementary Table 1.

-L156- nutrient starvation conditions (H₂O)

-Autophagy flux cycling- was the native promoter of ATG8 used for this? If not, what promoter was used? GFP signal is really strong in the images. Please state it in the text. Moreover, the authors state only the N terminus of ATG8. Although authors give references according to this construct, please state the nature of the promoter and section of the ATG8 gene used for better comprehension when reading.

-Panel g from figure 3 for Drim15 mutant is very difficult to see. Please enlarge it.

- L208- typo error: change then for than: less abundant than

Response to Reviewer Comments

Reviewer #4 (Remarks to the Author):

Minor

- Now that the residue S24 has been identified, is there any lead/evidence/experiment to show that Rim15 is the upstream kinase required for the phosphorylation at that residue? Is this residue conserved in other fungi?

We thank the Reviewer for this suggestion, which motivated us to look more closely at the consensus motif targeted by Rim15 in yeast, which Dokládál et al., 2021, determined was (S/T)P. We then examined the sequence adjacent to Ser24 in *M. oryzae* Gdh2 and found that it is followed by Pro25. Thus, Gdh2 carries the SP kinase consensus motif for Rim15. This information has been added to the text at line 224. Therefore it seems likely that, like in other fungi, the Rim15 consensus motif is (S/T)P, and this motif is conserved in *M. oryzae* Gdh2.

- In the model (Fig 9) it will be misleading to show with a continuous arrow the relationship between Rim15 kinase which will imply (could be interpreted as) direct phosphorylation. Include a discontinued arrow line.

A discontinued line has been added as suggested from Rim15 to autophagy.

-In the supplementary table 1, no MGG numbers are shown. It is essential that around L201 where Gdh2, Gdh1 and Glt1 are identified, that UniProt codes are given to be able to easily identify them on the Supplementary Table 1.

In both Supplementary Table 1 and 2, MGG numbers are included at the end of the information provided in the "Description" column of both tables, and also in the second from last "Ensemble Gene ID" column in Supplementary Table 1. However, we have added UniProt codes to the main text as suggested to enable easy identification in the tables (lines 208-215).

-L156- nutrient starvation conditions (H2O)

Done

-Autophagy flux cycling- was the native promoter of ATG8 used for this? If not, what promoter was used? GFP signal is really strong in the images. Please state it in the text. Moreover, the authors state only the

N terminus of ATG8. Although authors give references according to this construct, please state the nature of the promoter and section of the ATG8 gene used for better comprehension when reading.

Thank you for these comments. We have now made clear in the main text and in the Materials section that the full-length *ATG8* gene, fused to *GFP*, is under RP27 control (lines 154-155 and lines 430-435).

-Panel g from figure 3 for Drim15 mutant is very difficult to see. Please enlarge it.

Done

- L208- typo error: change then for than: less abundant than

Done